# Ankyrin-G mediates targeting of both Na$^+$ and K$_{ATP}$ channels to the rat cardiac intercalated disc

**Hua-Qian Yang[1], Marta Pérez-Hernández[2], Jose Sanchez-Alonso[3], Andriy Shevchuk[4], Julia Gorelik[3], Eli Rothenberg[5], Mario Delmar[2,6], William A Coetzee[1,5,7]\***

[1]Pediatrics, NYU School of Medicine, New York, United States; [2]Medicine, NYU School of Medicine, New York, United States; [3]National Heart and Lung Institute, Imperial Centre for Translational and Experimental Medicine, Imperial College London, London, United Kingdom; [4]Department of Medicine, Imperial College London, London, United Kingdom; [5]Biochemistry and Molecular Pharmacology, NYU School of Medicine, New York, United States; [6]Cell Biology, NYU School of Medicine, New York, United States; [7]Neuroscience and Physiology, NYU School of Medicine, New York, United States

**Abstract** We investigated targeting mechanisms of Na$^+$ and K$_{ATP}$ channels to the intercalated disk (ICD) of cardiomyocytes. Patch clamp and surface biotinylation data show reciprocal downregulation of each other's surface density. Mutagenesis of the Kir6.2 ankyrin binding site disrupts this functional coupling. Duplex patch clamping and Angle SICM recordings show that I$_{Na}$ and I$_{KATP}$ functionally co-localize at the rat ICD, but not at the lateral membrane. Quantitative STORM imaging show that Na$^+$ and K$_{ATP}$ channels are localized close to each other and to AnkG, but not to AnkB, at the ICD. Peptides corresponding to Nav1.5 and Kir6.2 ankyrin binding sites dysregulate targeting of both Na$^+$ and K$_{ATP}$ channels to the ICD, but not to lateral membranes. Finally, a clinically relevant gene variant that disrupts K$_{ATP}$ channel trafficking also regulates Na$^+$ channel surface expression. The functional coupling between these two channels need to be considered when assessing clinical variants and therapeutics.

**\*For correspondence:**
william.coetzee@nyu.edu

**Competing interests:** The authors declare that no competing interests exist.

## Introduction

Voltage-gated Na$^+$ channels are responsible for the initiation and propagation of action potentials in many excitable cell types, including neurons, skeletal muscle and cardiac myocytes. The pore-forming α-subunit of the cardiac Na$^+$ channel (Nav1.5) is encoded by the *SCN5A* gene. A large amount of genetic information has linked *SCN5A* variants to inherited forms of arrhythmias and sudden death, including Brugada syndrome, sick sinus syndrome, Long-QT syndrome and others (*Veerman et al., 2015*). Nav1.5 interacts with several types of proteins, including 14-3-3, Ca$^{2+}$/cal-modulin-dependent protein kinase II (CaMKII), Fibroblast growth factor 13 (FGF13), Ankyrin-G (AnkG) and several others (*Shy et al., 2013*). Mutations in these interactors are also associated with arrhythmogenic syndromes since they affect the Na$^+$ channel (*Shy et al., 2013*). It is of paramount importance, therefore, to know which proteins associate with Na$^+$ channels and how they affect Na$^+$ channel expression and function.

The sarcolemmal ATP-sensitive K$^+$ (K$_{ATP}$) channel is one of the most abundant channels expressed in cardiac myocytes and it promotes action potential shortening adaptation with elevated heart rates (*Foster and Coetzee, 2016*). K$_{ATP}$ channels additionally have important protective effects during metabolic stress and hypoxia/ischemia. Studies with murine genetic models have demonstrated that

sarcolemmal $K_{ATP}$ channels mediate a key component of the protective effects of ischemic preconditioning (*Foster and Coetzee, 2016*). As sensors of intracellular nucleotides (ATP, MgADP and AMP), $K_{ATP}$ channels couple alterations in energy metabolism to $K^+$ fluxes and membrane excitability (*Foster and Coetzee, 2016*). Intracellular ATP blocks the channel by binding to a pocket formed by the intracellular N- and C-termini of Kir6.x, whereas ADP promotes channel opening by binding to intracellular nucleotide binding folds on the partner subunit, SURx. Two genes (*KCNJ8* and *KCNJ11*) respectively code two distinct pore-forming Kir6.1 and Kir6.2 subunits, and two genes (*ABCC8* and *ABCC9*) code for the accessory SUR1 and SUR2 subunits. Two major SUR2 isoforms (SUR2A and SUR2B) exist as a result of alternative mRNA splicing (*Foster and Coetzee, 2016*).

Despite the obvious functional differences in cardiac $Na^+$ channels and $K_{ATP}$ channels, there are also similarities, particularly in their subcellular expression profiles. At least two distinct pools of $Na^+$ channels have been identified in cardiac myocytes (*Shy et al., 2013*). One pool is targeted to lateral membranes by the syntrophin/dystrophin complex (*Gavillet et al., 2006*; *Hund et al., 2010*), whereas another subpopulation is organized in a highly specialized macromolecular complex at the intercalated disk (ICD) region (*Agullo-Pascual et al., 2014*), where targeting and anchoring is coordinated by AnkG, Synapse-associated protein 97 (SAP97), Microtubule plus-end binding protein (EB1) and Plakophilin-2 (PKP2) (*Shy et al., 2013*; *Gillet et al., 2014*). $K_{ATP}$ channel subcellular expression follows a similar trend: The presence of $K_{ATP}$ channels in lateral membranes was established by their initial identification with patch clamp methods (*Foster and Coetzee, 2016*). However, as with $Na^+$ channels, $K_{ATP}$ channels are enriched at the ICD of cardiac myocytes where they morphologically cluster with desmosomal proteins such as PKP2 (*Hong et al., 2012*), suggesting that $K_{ATP}$ channels form part of an ICD channel/transporter complex that is gaining increasing recognition for roles in cell-cell communication and cell adhesion.

It is a common theme in cardiac electrophysiology that ion channels in the same subdomain have the potential to interact with each other, as well as with other channels and transporters. Here, we provide evidence that $Na^+$ channels and $K_{ATP}$ channels are morphologically clustered (particularly at the ICD) and that they interact functionally, most likely due to the fact that they are targeted to a common subcellular location by AnkG. These studies provide a new paradigm when considering pharmacological and genetic aspects of arrhythmogenesis.

## Results

### $K_{ATP}$ and Nav1.5 channels negatively regulate each other in HEK293 cells

Distinct subpopulations of $Na^+$ channels and $K_{ATP}$ channels exist in a cardiomyocyte, but the relevance of this observation has been unclear. Recent studies have shown that the function and trafficking of $Na^+$ channels can be regulated by some $K^+$ channels, including Kir2.1 and Kv4.3 (*Portero et al., 2018*; *Ponce-Balbuena et al., 2018*). We investigated whether an interaction exists between $Na^+$ and $K_{ATP}$ channels. First, we performed experiments with HEK293 cells transfected with only $K_{ATP}$ channels (Kir6.2/SUR2A), only $Na^+$ channels (Nav1.5), both type of channels, or with an empty vector (pcDNA3) as a negative control. Unlike Kir2.1, which positively regulates Nav1.5, we unexpectedly found that the whole-cell $Na^+$ current density was significantly smaller when $K_{ATP}$ channels were present (*Figure 1A and B*). The voltage-dependence of steady-state activation of the $Na^+$ channel and the inactivation kinetics were unchanged by the presence of $K_{ATP}$ channels (*Figure 1C* and *Figure 1—figure supplement 1*). We next tested whether $K^+$ flux through $K_{ATP}$ channels plays a role. To answer this question, we took advantage of the fact that mutating the GFG sequence in the Kir6.2 pore to AAA produces a non-functional channel that still traffics normally to the cell membrane (*Tong et al., 2006*). As with the wild-type channel, Kir6.2-AAA also suppressed Nav1.5 currents (*Figure 1—figure supplement 2*), demonstrating that a conducting $K_{ATP}$ channel is not required. Reciprocal functional interaction occurs since the $K_{ATP}$ channel mean patch current in excised patches were significantly reduced when $Na^+$ channels were present (*Figure 1D and E*). The presence of $Na^+$ channels did not affect the sensitivity of $K_{ATP}$ channels to 'cytosolic' ATP (*Figure 1F*). Of note, the negative regulation of one channel by the other was not due to differences in transfection efficiency, transcription or translation since the Nav1.5 protein levels in cell lysates were unchanged in the presence of $K_{ATP}$ channels, and co-transfection with Nav1.5 did not affect

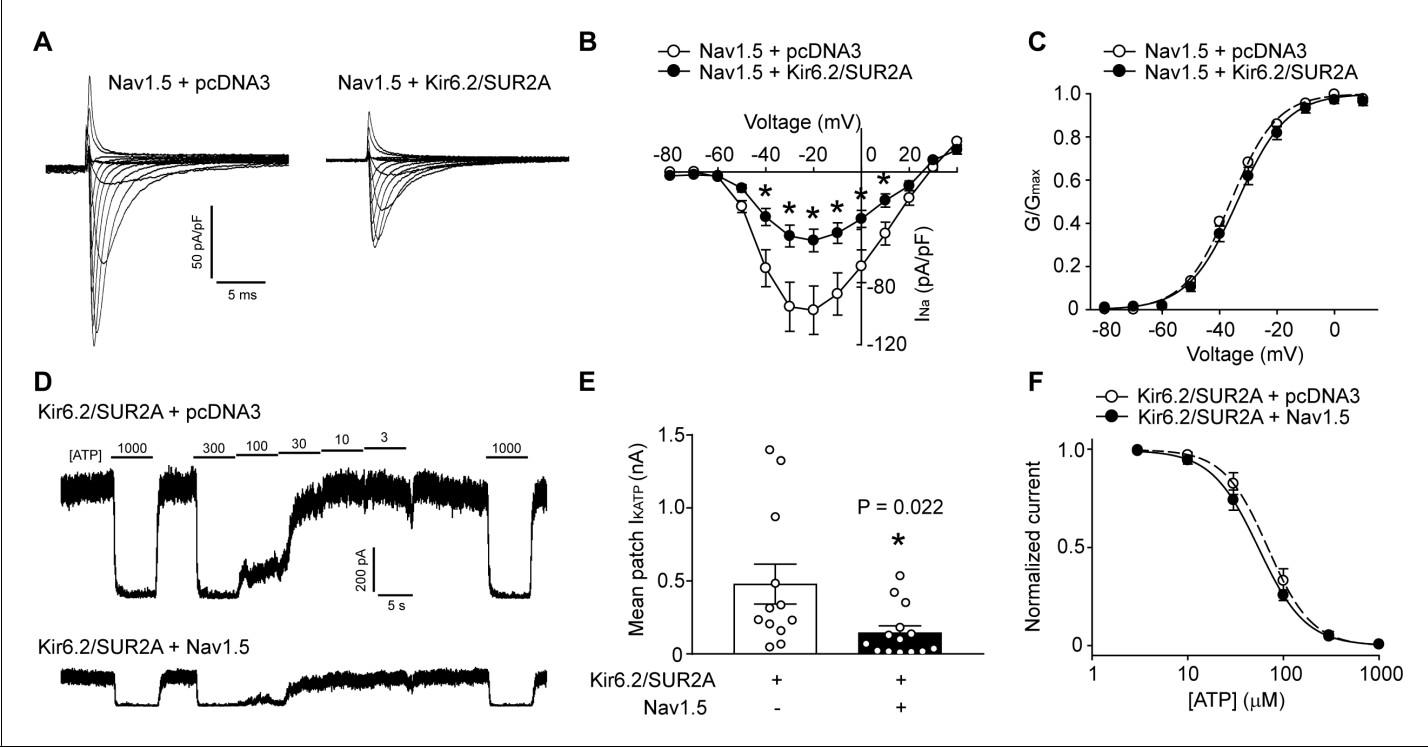

**Figure 1.** $K_{ATP}$ channels and $Na^+$ channels reciprocally reduce the functional expression of one another. (**A**) Representative whole-cell $Na^+$ current recordings from Nav1.5 transfected HEK293 cells co-transfected with empty vector (pcDNA3) or $K_{ATP}$ channel (Kir6.2+SUR2A). (**B**) Averaged current-voltage relationships of Nav1.5 co-expressed with empty vector (open symbols; n = 12) or Kir6.2/SUR2A (filled symbols; n = 10). *p<0.001 determined by two-way ANOVA followed by Tukey's test. (**C**) The voltage dependence of steady-state activation was calculated from the traces in panel A. Values of $G_{Na}$ were normalized to the maximum conductance and plotted as a function of voltage. The symbols have the same meaning as in panel B. (**D**) Representative inside-out current recordings obtained from $K_{ATP}$ channel (Kir6.2+SUR2A) transfected HEK293 cells co-transfected with empty vector (pcDNA3) or Nav1.5. ATP concentrations (µM) were switched as indicated. The mean patch current was recorded at a membrane potential of −80 mV (a voltage at which Nav1.5 is inactive) and the $K_{ATP}$ channel current was defined by the current component blocked by 1 mM ATP applied to the 'cytosolic' face of the patch. Recordings were made immediately after patch excision to minimize effects of 'run-down'. (**E**) Data points from Kir6.2/SUR2A transfected cells co-transfected with empty vector (pcDNA3) (open symbols; n = 12) or Nav1.5 (filled symbols; n = 14). *p=0.022 using the Student's *t* test. (**F**) The ATP-sensitivity of $K_{ATP}$ channels was determined by plotting the $K_{ATP}$ current (normalized to the maximum current) as a function of the 'cytosolic' ATP concentration. Data from individual patches were subjected to curve fitting to a modified Boltzmann equation, yielding $IC_{50}$ values for ATP inhibition of 63.0 ± 9.5 µM and 66.2 ± 10.6 µM respectively for Kir6.2/SUR2A without and with Nav1.5. Data are from a minimum of 3 separate transfections.

The online version of this article includes the following figure supplement(s) for figure 1:

**Figure supplement 1.** Co-expression with $K_{ATP}$ channels does not affect Nav1.5 channel inactivation.

**Figure supplement 2.** Non-conducting $K_{ATP}$ channels negatively regulate Nav1.5.

total Kir6.2 protein levels (*Figure 2*). Collectively, these data demonstrate that $K_{ATP}$ channels and $Na^+$ channels negatively regulate each other's function when overexpressed in a heterologous expression system.

## Negative regulation is imparted by a reduction in surface expression

Since the current densities were reduced by co-expression, but other channel properties remained unaltered, we used surface biotinylation assays to investigate whether surface expression was reduced. Data from these experiments demonstrated that the surface expression of Nav1.5, relative to the total Nav1.5 protein in the cell lysates, was significantly reduced when $K_{ATP}$ channels were co-expressed (*Figure 2A and B*). To assess $K_{ATP}$ channel surface expression we used a Kir6.2 construct with an extracellular Avi tag and C-terminal myc tag (Avi-Kir6.2-myc), which functions and traffics similar to wild-type Kir6.2 (*Yang et al., 2018*). Similar to the previous result, we found that $K_{ATP}$

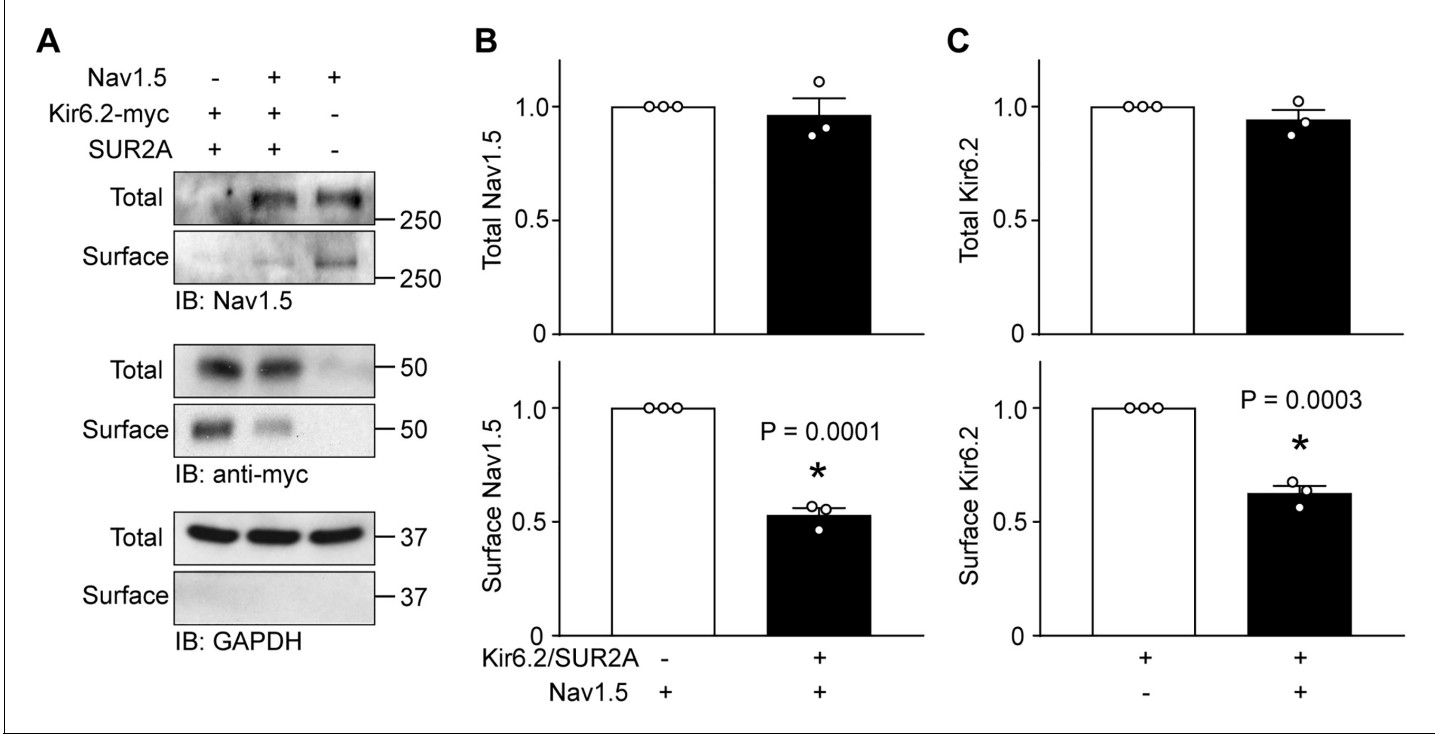

**Figure 2.** $K_{ATP}$ channels and $Na^+$ channels reciprocally reduce the surface expression of each other. HEK-293 cells were transfected with combinations of Kir6.2 (C-terminal tagged with 6 × myc epitopes), SUR2A, Nav1.5 as indicated. pcDNA3 was included to keep the cDNA amounts equivalent in transfections. (**A**) Cell lysates (Total) or surface biotinylated membrane fractions (Surface) were subjected to SDS-PAGE and immunoblotted with antibodies against Nav1.5, myc, or GAPDH. A representative immunoblot is shown. Panels B and C respectively show data averaged from three similar blots. Total Nav1.5 or Kir6.2 protein in cell lysates were normalized to the amount of GAPDH, whereas surface expression was normalized to the total Nav1.5 or Kir6.2 protein. *p=0.0001 and p=0.0003 for panel B and C respectively with Student's $t$ test.

channel surface expression was significantly impaired by $Na^+$ channels overexpression (**Figure 2A and C**).

## A key role for the Kir6.2 ankyrin binding site

The surface abundance of membrane proteins can be regulated by anchoring mechanisms. $Na^+$ and $K_{ATP}$ channels have been reported to be respectively regulated by Ankyrin-G and B (**Mohler et al., 2004**; **Kline et al., 2009**). The AnkG binding motif of Nav1.5 consists of amino acids VPIAVAESD (**Mohler et al., 2004**), whereas the Kir6.2 C-terminal amino acid sequence VPIVAEED is necessary for in vitro binding to a GST-tagged AnkB membrane-binding domain, AnkB-MBD (**Figure 3A**) (**Kline et al., 2009**). Mutagenesis of negatively charged amino acids within this motif (E321K, E322K and D323K) disrupts both Kir6.2/AnkB-MBD binding and Kir6.2 surface expression (**Kline et al., 2009**). To investigate the potential role of ankyrin binding, we mutated the EED residues to lysine (Kir6.2-KKK). These mutations decreased Kir6.2 surface expression in HEK293 cells and we had to increase the cDNA amounts during transfections to accomplish cellular protein levels comparable to wild-type (**Figure 3—figure supplement 1**). As before, co-expression with WT-Kir6.2/SUR2A significantly reduced whole-cell Nav1.5 currents. By contrast, Kir6.2-KKK/SUR2A did not functionally interact with $Na^+$ channels (**Figure 3B**). Moreover, surface biotinylation data demonstrated that WT-Kir6.2/SUR2A reduced surface expression of Nav1.5 as expected, but Kir6.2-KKK/SUR2A had no such effect (**Figure 3C and D**). We investigated whether the functional interaction also occurs with the other member of the Kir6 subfamily, namely Kir6.1, which does not bind to AnkB-MBD (**Kline et al., 2009**). Interestingly, Kir6.1/SUR2A was without effect on Nav1.5 currents (**Figure 3B**). Note that SUR2A was also present in this experiment, which supports the argument that the phenotype is intrinsic to Kir6.2, and not to SUR2A. Overall, these results demonstrate that the functional

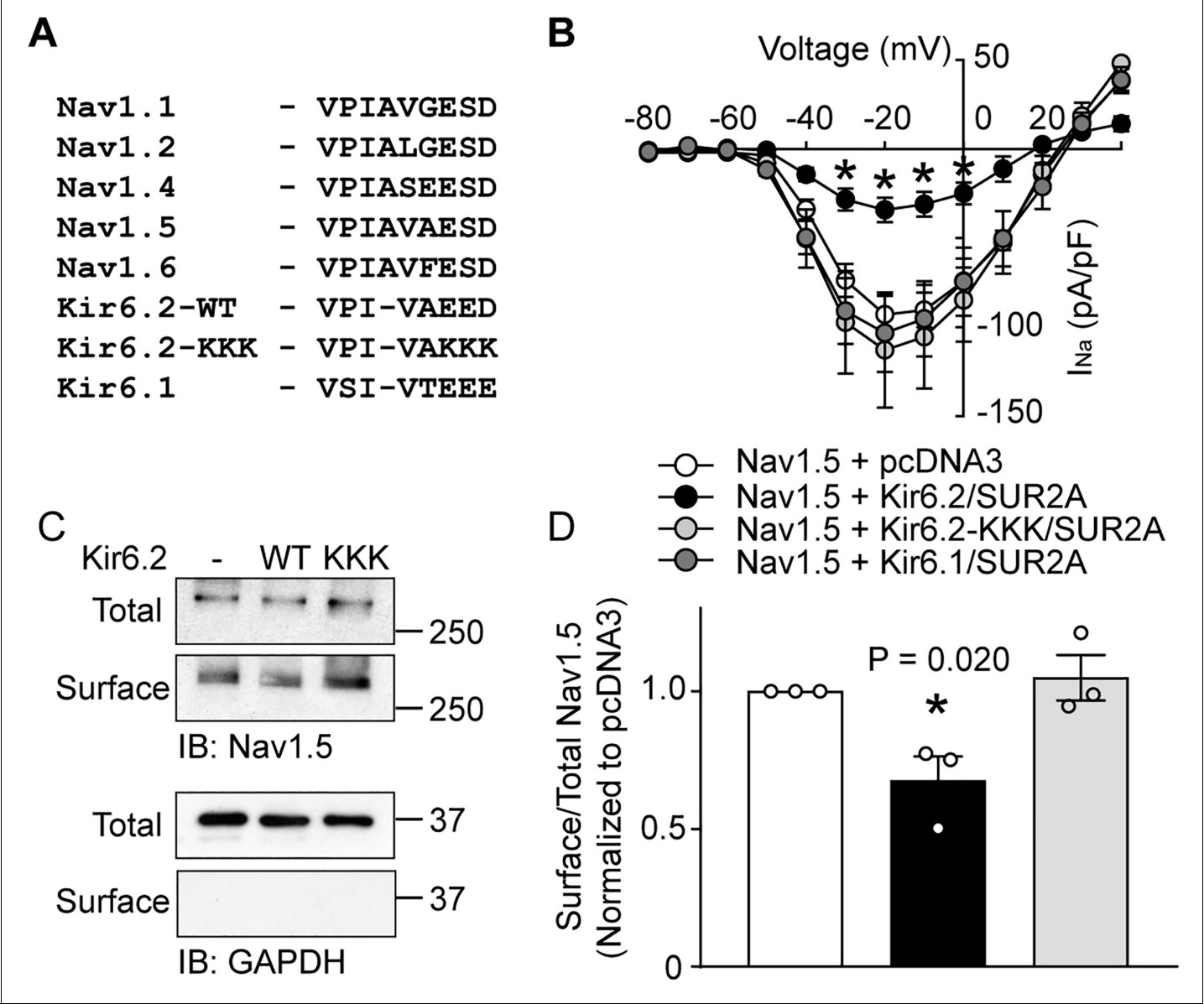

**Figure 3.** An intact Kir6.2 ankyrin binding motif is necessary for functional interaction. (A) Sequence alignment of ankyrin binding motifs (or corresponding residues) of various Nav channels and members of the Kir6 subfamily of inward rectifier K+ channels. The Kir6.2-KKK shows the mutations made to disrupt the binding motif. (B) Whole-cell Na+ currents were measured in HEK293 cells and averaged current-voltage relationships are plotted for cells transfected with Nav1.5 plus either empty vector (n = 8), Kir6.1/SUR2A (n = 6), Kir6.2/SUR2A (n = 7) or Kir6.2-KKK/SUR2A (n = 6). Data are from a minimum of 3 transfections. *p<0.001 determined by two-way ANOVA followed by Tukey's test. (C) HEK-293 cells were transfected with Nav1.5 plus combinations of empty vector, Kir6.2/SUR2A, or Kir6.2-KKK/SUR2A. Cell lysates (Total) or surface biotinylated membrane fractions (Surface) were subjected to SDS-PAGE and immunoblotted with antibodies against Nav1.5 or GAPDH. A representative immunoblot is shown. (D) Averaged data of Nav1.5 surface expression normalized to the total Nav1.5 protein from three similar blots. *p=0.020 with 1W ANOVA followed by Dunnett's test. The online version of this article includes the following figure supplement(s) for figure 3:

**Figure supplement 1.** Titration of the cDNA amounts used in transfection reactions to attain similar cell protein amounts.

interactions of Nav1.5 and $K_{ATP}$ channels are tightly coupled to the presence of an intact ankyrin binding domain in Kir6.2, therefore suggesting the possibility that competition for ankyrin binding mediates the functional interaction.

# Functional co-localization of Na$^+$ and K$_{ATP}$ channels at the intercalated disc region of cardiac myocytes

We started to investigate whether Na$^+$ channels and K$_{ATP}$ channels functionally interact in cardiomyocytes by overexpressing Kir6.2 with a C-terminal mEos tag (Ad.Kir6.2-mEos) via adenoviral delivery. Indeed, rat cardiomyocytes expressing Kir6.2-mEos had a significantly reduced whole-cell Na$^+$ current density compared to Ad.mCherry as a negative control (*Figure 4—figure supplement 1*). A key question, however, is whether the functional co-localization of Na$^+$ channels and K$_{ATP}$ channels can be demonstrated at a subcellular level. To accomplish this goal, we have developed a novel duplex patch clamp technique that allows sequential measurements of the two currents in the same

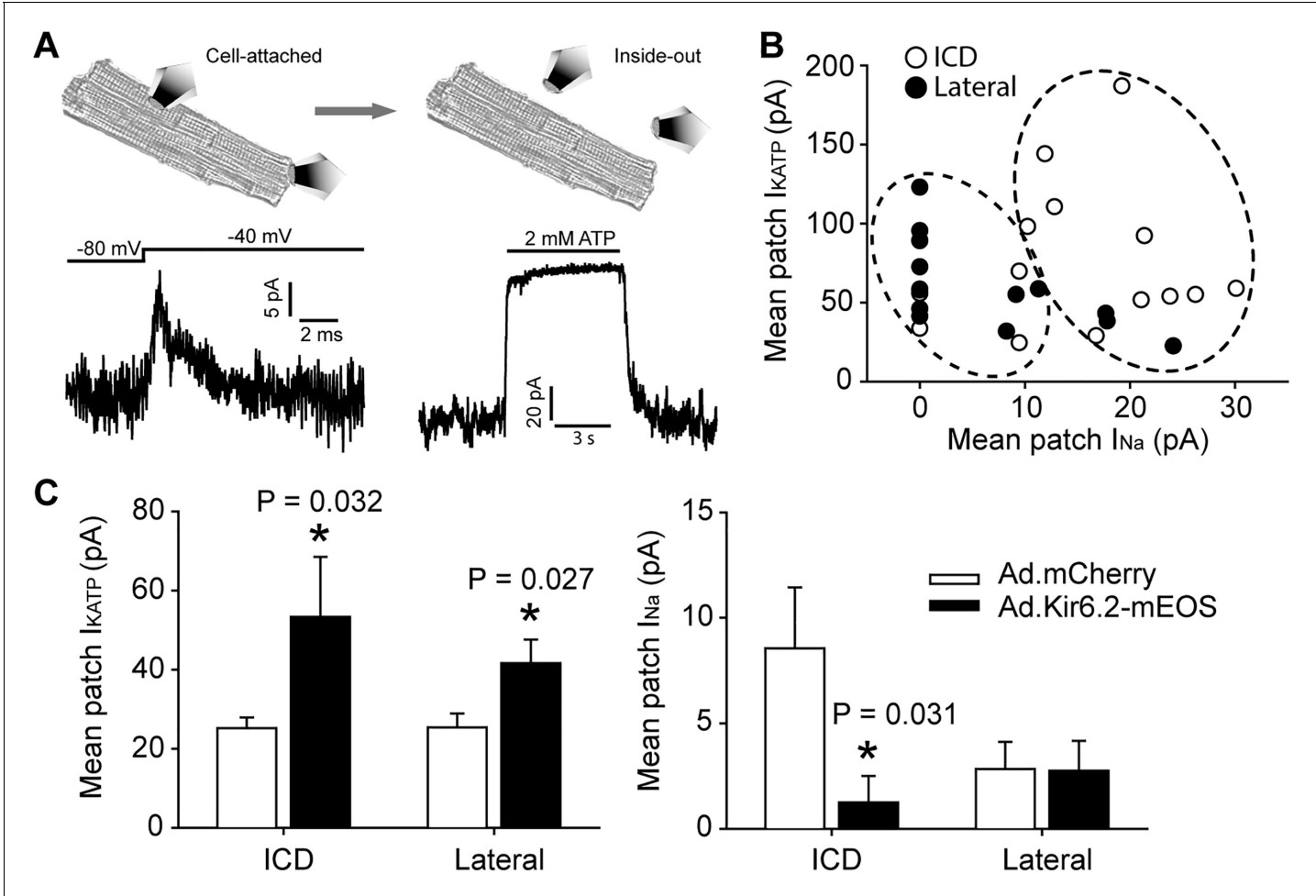

**Figure 4.** Functional interaction between K$_{ATP}$ channels and Na$^+$ channels in cardiac myocytes occurs predominantly at the ICD. (**A**) Illustration of the duplex patch clamp method, which allows sequential recordings of I$_{Na}$ and I$_{KATP}$ in the same membrane patch. In cell-attached mode (left), I$_{Na}$ is recorded by changing the membrane potential. Recordings can be made in the absence of I$_{KATP}$, which is generally not active in resting cardiomyocytes. After patch excision (right), recordings of K$_{ATP}$ channels are made at +80 mV, a voltage at which I$_{Na}$ is inactivated. Representative patch recordings are illustrated. (**B**) Paired duplex patch clamp recordings of I$_{KATP}$ and I$_{Na}$ are plotted as individual points for patches from the lateral membrane (filled symbols; n = 15) or the ICD region of cardiomyocytes (open symbols; n = 14). The dotted circles illustrate the result of an independent hierarchal clustering analysis of these data (*Figure 4—figure supplement 2*). Data were obtained using cells from three separate isolations. (**C**) Duplex patch clamping was performed with rat cardiomyocytes treated with Ad.mCherry or Ad.Kir6.2-mEos. The average I$_{KATP}$ or I$_{Na}$ recorded from lateral membranes (n = 14 for Ad.mCherry and 11 for Ad.Kir6.2-mEos) or the ICD (n = 13 for Ad.mCherry and 10 for Ad.Kir6.2-mEos) are plotted as bar graphs. Data were obtained using cells from four separate isolations. *p=0.032 and p=0.027 respectively for left panel, p=0.031 for right panel vs. Ad-mCherry using the Student's *t* test.

The online version of this article includes the following figure supplement(s) for figure 4:

**Figure supplement 1.** Adenoviral delivery of Kir6.2 in adult cardiomyocytes reduces whole-cell Nav1.5 current.

**Figure supplement 2.** Hierarchical clustering of duplex patch clamp data.

membrane patch (illustrated in *Figure 4A*). Assuming that the free patch area of a 2–3 MΩ pipette is ~10 μm$^2$ (*Sakmann and Neher, 1995*), this technique has a spatial resolution of ~2–3 μm. The $I_{Na}$ is first recorded in a cell-attached membrane patch as in our previous studies (*Lin et al., 2011*). This Na$^+$ channel recording is uncontaminated by $K_{ATP}$ channels, which are closed at rest in an intact cell. The patch is then excised to measure the $I_{K(ATP)}$ mean patch current in an inside-out membrane patch. The membrane voltage is kept at +80 mV to inactivate $I_{Na}$ and the magnitude of $I_{K(ATP)}$ is defined as the current component that is blocked by 'intracellular' ATP. Such paired recordings were made at the lateral membrane, or as close as feasible to the end of the cardiomyocyte (the ICD region) and paired recordings are summarized in *Figure 4B*. A key observation was that the majority of ICD patches expressed $I_{Na}$, but it was found in only ~50% of lateral patches. By contrast, each of the patches from lateral membranes and the ICD contained $K_{ATP}$ channels. These paired data (blinded to their origin) were analyzed with machine learning algorithms (hierarchical clustering), which demonstrated two distinct populations (*Figure 4—figure supplement 2*). The lateral and ICD paired recordings segregated statistically between these two clusters (*Figure 4B*; Fisher's Exact test, p=0.002), demonstrating that the ICD and lateral paired data points are distinct populations. To examine functional co-localization of Na$^+$ channels and $K_{ATP}$ channels at a subcellular level, we next performed duplex patch clamping after adenoviral delivery of Kir6.2-mEos. This intervention led to significantly larger $K_{ATP}$ channel currents at both the lateral membranes and at the ICD. The corresponding $I_{Na}$ patch current was significantly reduced at the ICD, but not at lateral membranes (*Figure 4C*). Thus, cardiac Na$^+$ channels and $K_{ATP}$ channels functionally interact predominantly at the ICD in cardiac myocytes.

To investigate the functional co-localization of Na$^+$ and $K_{ATP}$ channels at the ICD with improved spatial resolution, we performed Angle SICM experiments with high resistance (~30 MΩ) patch pipettes. With this methodology, a topographical image of the ICD surface is produced before performing cell attached patch clamping at a selected position (*Figure 5A and B*). The Na$^+$ channels were activated by voltage clamp steps from −120 mV, whereas $K_{ATP}$ channels were simultaneously recorded by activating them with pinacidil included in the pipette solution. Similar to our previous report (*Leo-Macias et al., 2016*), around 80% of recordings from ICD did not show Na$^+$ channels activity. Notably, the patches without $I_{Na}$ also lacked $K_{ATP}$ channels. However, the Angle SICM patches from the ICD that contained a cluster of Na$^+$ channels also contained $K_{ATP}$ channel activity (*Figure 5C and D*). Thus, these data further support the concept that Na$^+$ channels and $K_{ATP}$ channels are functionally co-localized at the ICD of cardiac myocytes.

## $K_{ATP}$ channels colocalize with AnkG but not AnkB at intercalated disc

In cardiomyocytes, AnkG localizes primarily, but not exclusively, to the ICD (*Mohler et al., 2004*; *Lowe et al., 2008*) and targets Nav1.5 to the ICD in cardiomyocytes (*Mohler et al., 2004*; *Makara et al., 2014*; *Knezl et al., 2008*). By contrast, AnkB is expressed mainly at lateral membranes where it localizes to Z- and M-lines in an isoform-dependent manner (*Wu et al., 2015*). Although Kir6.2 can interact with an AnkB-MBD construct (*Kline et al., 2009*), co-localization of AnkB and $K_{ATP}$ channels has not been investigated in adult cardiomyocytes. Given the prominent role for the Kir6.2 ankyrin binding site in the functional interaction between $K_{ATP}$ channels and Na$^+$ channels, we next asked whether $K_{ATP}$ channels co-localize with Ankyrins (AnkG or AnkB). Validation of anti-Nav1.5 and anti-Kir6.2 antibodies is shown in *Figure 6—figure supplement 1*. Immunofluorescence confocal microscopy of isolated rat cardiomyocytes and cardiac cryosections confirmed the presence of AnkB at lateral membranes, whereas expression of AnkG is enriched at (but not restricted to) the ICD (*Figure 6—figure supplement 2*). We found little evidence for co-localization of Kir6.2 with AnkB. By contrast, consistent with our previous report (*Hong et al., 2012*), we found that Kir6.2 co-localizes with AnkG, particularly at the ICD region of cardiac myocytes (*Figure 6—figure supplement 2*). We used STORM super-resolution microscopy to better analyze and quantify the co-localization of Nav1.5 and Kir6.2 with ankyrins. *Figure 6* shows images of rat cardiomyocyte intercalated disc regions, co-stained with Kir6.2 and AnkB or AnkG, as well as Nav1.5 co-stained with AnkG. We quantified the distances between cluster edges of channels and ankyrins in the ICD region, which demonstrated a median clustering distance of 150 nm (interquartile range: 0–433 nm) between Nav1.5 and AnkG, which are well characterized to co-localize and interact at the ICD (*Mohler et al., 2004*; *Makara et al., 2014*; *Knezl et al., 2008*). The clustering distance between Kir6.2 and AnkG at the ICD was in the same range (median: 228 nm, interquartile range: 0–563 nm),

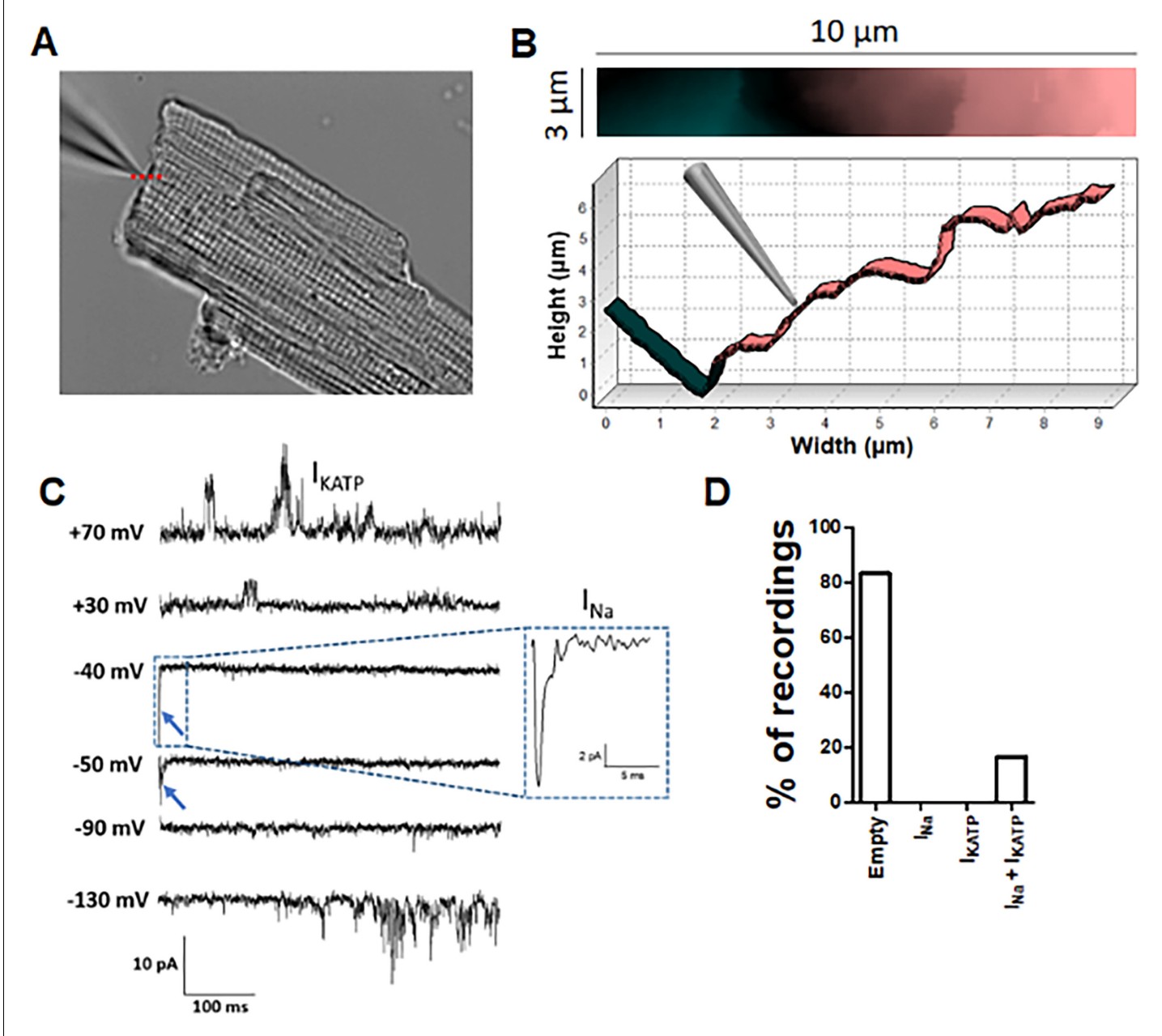

**Figure 5.** Angle SICM recordings demonstrate functional co-localization of Na$^+$ and K$_{ATP}$ channels at the ICD of adult rat ventricular cardiomyocytes. (A) Representative phase contrast image of a single cardiomyocyte. Recording pipette can be observed in the upper left. (B) Scan image acquired from location marked with red dashed line in A showing ICD region (top panel). Cross section showing the position of the pipette with respect the ICD (bottom). (C) Representative traces of the recorded current. Na$^+$ channel currents can be observed at −40 mV and −50 mV steps (blue arrow and dashed box), K$_{ATP}$ channel currents can be observed at +70 mV and −130 mV, when the Na$^+$ channels are inactive. (D) Summary of cell-attached patch recordings. A total of 12 seals were recorded in which 10 of them did not show any channel activity at any voltage and 2 of them shown both Na$^+$ and K$_{ATP}$ channels activity.

whereas the clustering distance between Kir6.2 and AnkB was four times larger (median: 886 nm, interquartile range: 102–2205 nm). Thus, both Nav1.5 and Kir6.2 are strongly co-localized with AnkG, but not with AnkB, at the ICD of cardiac myocytes.

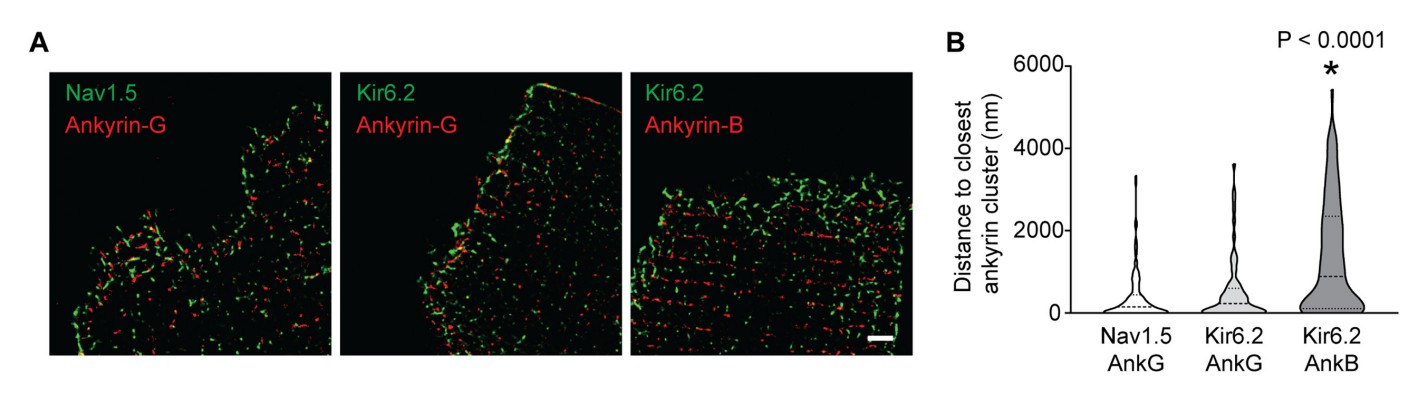

**Figure 6.** Quantification of inter-cluster distances between $K_{ATP}$ channels, $Na^+$ channels and ankyrins. (**A**) Enlarged STORM images of the rat cardiomyocyte intercalated disc region, co-stained with antibodies against ankyrin B, ankyrin G, Kir6.2 and Nav1.5, as indicated. Scale bar, 2 μm. (**B**) Statistical analysis of the distances between Nav1.5 or Kir6.2 to the closest ankyrin clusters. N ≥ 67 clusters from three rats in each group. *p<0.0001 vs. Nav1.5/AnkG group determined by Kruskal-Wallis 1W ANOVA, followed by Dunn's post-hoc analysis.

The online version of this article includes the following figure supplement(s) for figure 6:

**Figure supplement 1.** Validation of Nav1.5 and Kir6.2 antibodies for immunostaining in cardiomyocytes.

**Figure supplement 2.** Co-localization of $K_{ATP}$ channels, $Na^+$ channels and Ankyrins at the ICD of cardiac myocytes.

**Figure supplement 3.** Co-localization of Kir6.2 and Nav1.5 at intercalated disc.

## Ankyrin binding is necessary to localize $Na^+$ and $K_{ATP}$ channels to the ICD

We next examined whether Ankyrin binding promotes trafficking of $Na^+$ channels and $K_{ATP}$ channels to the ICD of cardiac myocytes. To address this question, we used peptides corresponding to the Kir6.2 or Nav1.5 ankyrin binding sites to outcompete binding of the channels to ankyrins. The peptides were conjugated to an HIV Tat-derived peptide to enable delivery into cells (*Schwarze et al., 1999*; *Chen et al., 2001*). Rat cardiac myocytes were treated for 24 hr with TAT peptides and the $I_{Na}$ and $I_{KATP}$ were measured using the duplex patch clamp technique, either at lateral membranes or at the ICD region. Averaged data are shown in *Figure 7*, which demonstrates that the $I_{Na}$ and $I_{KATP}$ were larger at the ICD than at lateral membranes as expected. Peptides corresponding to the Nav1.5 ankyrin binding site caused a reduction of both $I_{Na}$ and $I_{KATP}$ at the ICD, but not in lateral membranes. Similarly, peptides corresponding to the Kir6.2 ankyrin binding site caused targeting defects of both $I_{Na}$ and $I_{KATP}$ at the ICD, but not in lateral membranes. These peptide experiments demonstrate that ankyrin binding regulates targeting of both $Na^+$ channels and $K_{ATP}$ channels to the ICD of cardiac myocytes.

## Clinical variants that causing $K_{ATP}$ channel trafficking defects affect Nav1.5 surface expression

Variants in the genes encoding Kir6.2 and SUR1, *KCNJ11* and *ABCC8*, are commonly associated with insulin secretion disorders and diabetes (*Nichols, 2006*) and many of these variants cause trafficking defects of $K_{ATP}$ channels. For example, the missense NM_000525.3(KCNJ11):c.776A > G variant, which is rare in gnomAD and ExAC databases and is associated with severe congenital hyperinsulinism (ClinVar ID 8677), causes a p.His259Arg amino acid change in Kir6.2, and a severe trafficking defect (*Yang et al., 2017*; *Marthinet et al., 2005*). Given the functional interaction between $K_{ATP}$ channels and $Na^+$ channels, the question arose whether a clinically relevant $K_{ATP}$ channel gene variant such as this would affect the cardiac $Na^+$ channel. To answer this question, we co-expressed Nav1.5 with either wild-type Kir6.2 or Kir6.2-H259R in HEK293 cells. The Kir6.2-H259R channels neither expressed functional channels in patch clamp assays, nor were detected as surface proteins with biotinylation assays (*Figure 8A and B*), which is consistent with the trafficking defect previously reported (*Marthinet et al., 2005*). As before, the whole-cell Nav1.5 current was substantially reduced by co-expression with wild-type Kir6.2/SUR2A (compare *Figures 1A* and *8D*). By contrast, co-expression with Kir6.2-H259R/SUR2A did not reduce the Nav1.5 currents relative to empty

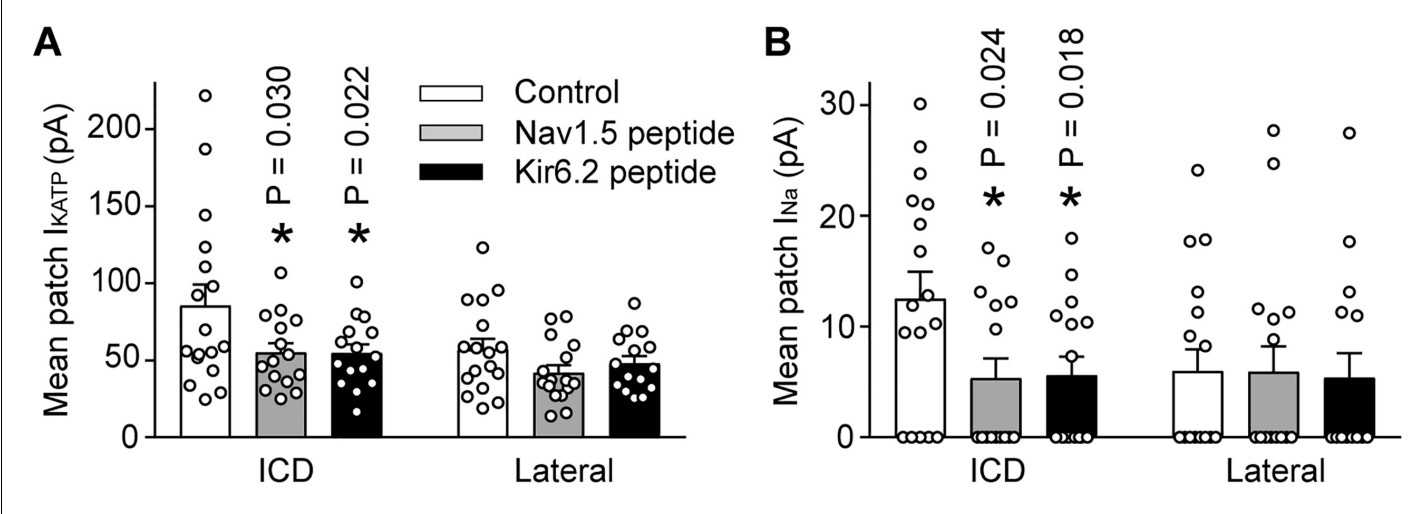

**Figure 7.** The $I_{KATP}$ and $I_{Na}$ at the ICD is preferentially reduced by peptides corresponding to the Nav1.5 or Kir6.2 ankyrin binding sites. Rat ventricular cardiac myocytes were incubated for 24 hr with TAT-conjugated peptides corresponding to the ankyrin binding site of Nav1.5 (Nav1.5 peptide, 50 µM; n = 15 cells) or Kir6.2 (Kir6.2 peptide, 50 µM; n = 15 cells). Untreated cardiomyocytes were used as a control (n = 17 cells). Duplex patch clamping was performed to measure $I_{KATP}$ (**A**) and $I_{Na}$ (**B**) paired recordings at lateral membranes or at the ICD. Shown are cumulative data obtained from cells isolated from four rats. *p=0.030 and p=0.022 respectively for panel A, p=0.024 and p=0.018 respectively for panel B vs. control determined by 1W ANOVA followed by Tukey's test.

The online version of this article includes the following figure supplement(s) for figure 7:

**Figure supplement 1.** Co-immunoprecipation of Kir6.2 with ankyrins.

vector controls (~100 pA/pF at −20 mV; *Figures 1A* and *8D*) and the surface abundance of Nav1.5 was significantly higher when co-expressed with Kir6.2-H259R compared to wild-type Kir6.2 (*Figure 8C*). Thus, variants that influence $K_{ATP}$ channel surface expression may simultaneously affect $Na^+$ channel surface expression and therefore have the potential to contribute to abnormalities in cardiac excitability and arrhythmias.

## Discussion

Our data demonstrate that $Na^+$ channels and $K_{ATP}$ channels are functionally coupled, both in heterologous expression systems and in cardiac myocytes. Ankyrin binding appears to underlie functional coupling since it can be disrupted by mutations in the Kir6.2 ankyrin binding site. In cardiomyocytes, $Na^+$ channels and $K_{ATP}$ channels co-localize both functionally, demonstrated by duplex patch clamping, and morphologically, specifically at the ICD. We found AnkG, but not AnkB, to be expressed at the ICD. Quantitative super-resolution microscopy shows similar clustering distances between AnkG and $Na^+$ channels or $K_{ATP}$ channels at the ICD. Competition experiments with peptides corresponding to Nav1.5 and Kir6.2 ankyrin binding sites dysregulate targeting of both $Na^+$ channels and $K_{ATP}$ channels to the ICD, but not to lateral membranes. Finally, we demonstrate that a clinically relevant gene variant that affect $K_{ATP}$ channel trafficking also affects $Na^+$ channel surface expression.

### Targeting of ion channels to discrete subcellular compartments

Many ion channels are compartmentalized within cells. For example, in the highly polarized kidney epithelial cell, channels and transporters can either target to the basolateral or apical membranes (*Stoops and Caplan, 2014*). In cardiac myocytes, ion channels and transporters are also targeted to specific subcellular compartments for optimal cellular function. The L-type $Ca^{2+}$ channel (LTCC), for example, is targeted to couplons adjacent to ryanodine receptors to allow for efficient excitation-contraction. Ion channels and transporters are often targeted to caveolae and lipid rafts. Caveolae are generally thought to be present at the mouth of t-tubules and/or the crests between t-tubules and may contain ion channels such as a specific population of LTCCs, Hyperpolarization activated

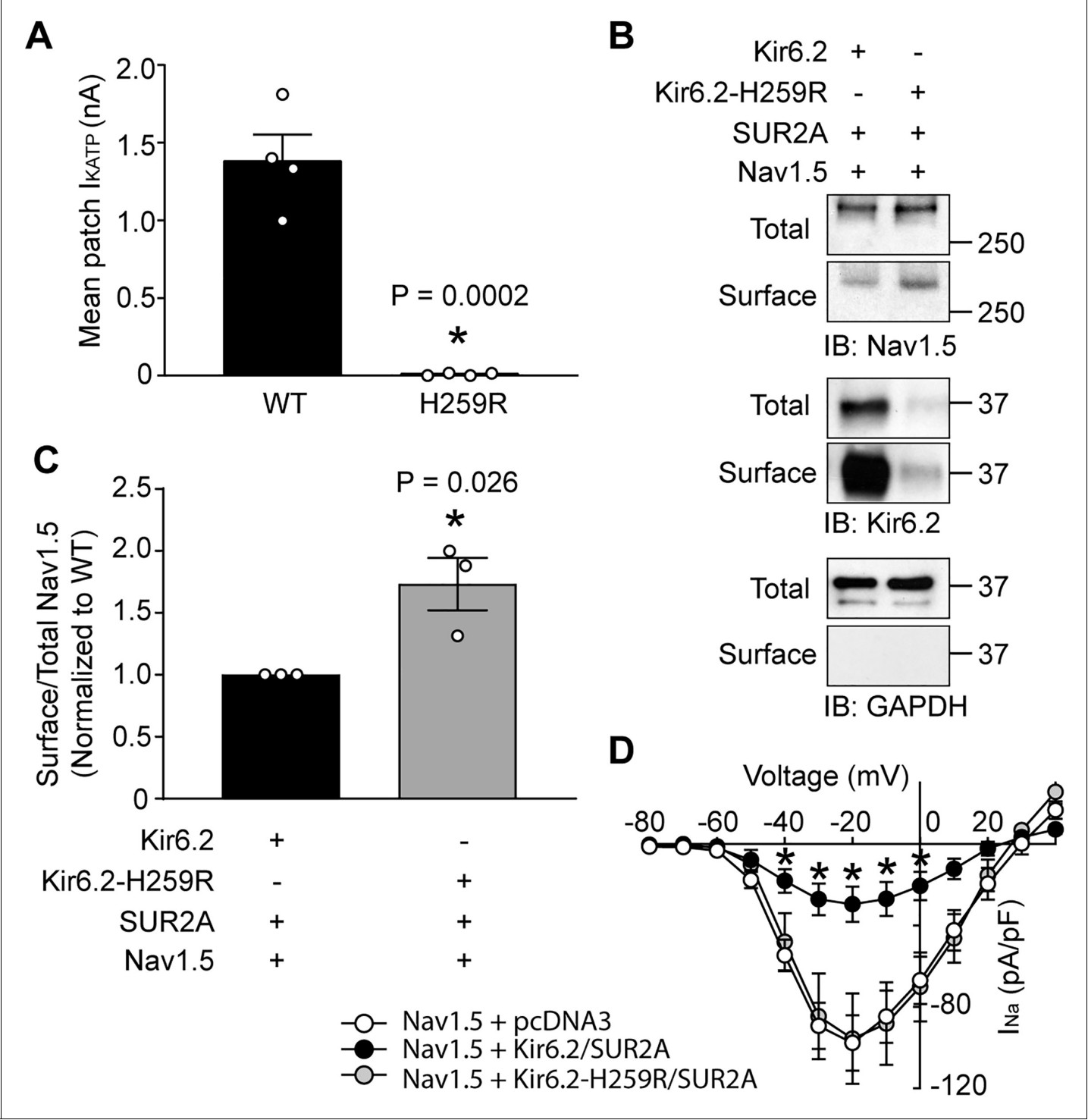

**Figure 8.** A clinically relevant Kir6.2 trafficking defective mutation (H259R) influences Nav1.5 surface expression. (**A**) HEK-293 cells were transfected with Kir6.2/SUR2A or Kir6.2-H259R/SUR2A. The averaged $K_{ATP}$ channel mean patch current, recorded in excised patches, are depicted as bar graphs, with WT-Kir6.2 and Kir6.2-H259R respectively depicted as black (n = 4 cells) or gray (n = 4 cells) bars. Data are from three transfections. *p=0.0002 using the Student's *t* test. (**B**) Surface expression was determined by biotinylation assays with cell lysates from HEK293 cells transfected with combinations of Nav1.5, Kir6.2, Kir6.2-H259R, and/or SUR2A. Shown is a representative immunoblot of cell lysates (total) or biotinylated membrane fractions (surface) probed with antibodies as indicated. (**C**) Summary data (n = 3) of the normalized ratio of surface/total Nav1.5 protein in cells transfected with cDNAs as indicated. *p=0.026 using the Student's *t* test. (**D**) Average current-voltage relationships of whole cell currents in HEK293 cells transfected with Nav1.5

*Figure 8 continued on next page*

*Figure 8 continued*

plus empty vector (n = 7), Kir6.2/SUR2A (n = 7) or Kir6.2-H259R/SUR2A (n = 7). Data are from three transfections. *p<0.05 determined by two-way ANOVA followed by Tukey's test.

The online version of this article includes the following figure supplement(s) for figure 8:

**Figure supplement 1.** Proposed model of Na$^+$ and K$^+$ flux coupling at the ICD.

cyclic nucleotide gated potassium channel 4 (HCN4), Kv1.5 and Na$^+$/Ca$^{2+}$ exchanger (*Best and Kamp, 2012*; *Hong and Shaw, 2017*). Caveolae are also enriched with components of molecular signaling pathways, such as β-adrenergic receptors, culminating in local regulation of ion channels. A number of ion channels are targeted to the ICD membrane structure that couple neighboring cardiac myocytes. The Na$^+$ channel was one of the first described to be enriched at the ICD (*Cohen and Levitt, 1993*). Examples of other channels enriched at the ICD of cardiac myocytes include Kv1.5 (*Mays et al., 1995*) and the inward rectifying K$^+$ channel subunits, Kir2.1 and Kir2.3 (*Melnyk et al., 2002*). In our studies, we found an enrichment of K$_{ATP}$ channels at the ICD regions of ventricular myocytes, where they co-localize with desmosomal proteins such as PKP2 (*Hong et al., 2012*). The targeting mechanisms for Na$^+$ channels to the ICD are best understood, and involve forwarding trafficking and anchoring mechanisms that are coordinated by EB1, SAP97, PKP2 and AnkG (*Shy et al., 2013*; *Agullo-Pascual et al., 2014*; *Gillet et al., 2014*; *Chen-Izu et al., 2015*). Mechanisms responsible for targeting K$_{ATP}$ channels to the ICD have not been described.

## A role for ankyrin G in targeting channels to the ICD

Ankyrins are cytoskeletal proteins that associate with spectrin-actin networks and bind to integral membrane proteins, thus serving as a sub-membrane scaffold for coordinating the targeting of membrane proteins. Axonal voltage-gated Na$^+$ channels, such as Nav1.6, has long been recognized to be targeted by AnkG (*Jenkins and Bennett, 2001*). Other ion translocators that are linked to the spectrin-based membrane skeleton by ankyrins include the anion exchanger, the Na$^+$/Ca$^{2+}$ exchanger, and the Na$^+$/K$^+$ ATPase (*Bennett and Baines, 2001*). A missense variant in *SCN5A* (the gene coding for Nav1.5), associated with Brugada syndrome, which causes an amino acid substitution (E1053K) in the Nav1.5 ankyrin-binding domain, has led to the identification of a key role for AnkG in targeting Na$^+$ channels to the ICD of cardiac myocytes (*Mohler et al., 2004*). Subsequent studies have shown that Na$^+$ channels are targeted to subcellular domains at the ICD that contain desmosomal proteins and N-cadherin (*Leo-Macias et al., 2016*; *Cerrone and Delmar, 2014*). K$_{ATP}$ channels are also enriched at the ICD and target to contain desmosomal proteins (*Hong et al., 2012*). Several observations from our current study suggest that AnkG plays a key role in K$_{ATP}$ channel targeting to this region. First, the functional interaction that exists between Na$^+$ channels and K$_{ATP}$ channels can be prevented by specific amino acid substitutions within the ankyrin binding domain of Kir6.2. Second, the quantitative STORM measurements show similar intermolecular distances between AnkG and Na$^+$ channels or K$_{ATP}$ channels. Third, duplex patch clamping shows functional co-localization in all patches obtained at the ICD between Na$^+$ channels and K$_{ATP}$ channels. Fourth, conventional fluorescence microscopy shows a remarkable degree of overlap between Nav1.5 and Kir6.2 staining at the ICD regions of cardiomyocytes. Fifth, competition experiments with peptides corresponding to Ankyrin binding motifs disrupt both Na$^+$ channel and K$_{ATP}$ channel expression at the ICD, demonstrating a clear role for ankyrins. Finally, consistent with the literature, we find strong expression of AnkG, but not AnkB, at the ICD. Collectively, we interpret these data as evidence that AnkG mediates targeting of both Na$^+$ channels and K$_{ATP}$ channels to the ICD.

Although we have observed the AnkG competition mechanism to regulate the surface expression of Nav1.5 and K$_{ATP}$ channels in both HEK293 cells and cardiomyocytes, a previous study has reported that the ankyrin binding deficient Nav1.5 E1053K efficiently transports to the membrane in HEK293 cells, but not in rat ventricular cardiomyocytes (*Mohler et al., 2004*). In the latter study, cells were co-transfected with the Nav-beta subunit, which was previously shown also to interact with AnkG (*Malhotra et al., 2002*). Thus, an additional role for Nav-beta subunits to regulate the AnkG/Nav1.5/K$_{ATP}$ complex is a real but unexplored possibility.

Both Na$^+$ and K$_{ATP}$ channels are enriched at the ICD of a cardiac myocyte (*Hong et al., 2012*; *Lin et al., 2011*). Previous data suggest that AnkG and Gap junction alpha-1 protein (Cx43) are

necessary to preserve the Na$^+$ current amplitude, electrical coupling and intercellular adhesion strength (*Sato et al., 2011*). Since the K$_{ATP}$ channel, Na$^+$ channel and AnkG interaction can be demonstrated in HEK-293 cells, which do not express Cx43, we believe that our data demonstrate that Cx43 is not a necessary component for functional interaction. However, we have not investigated this in cardiomyocytes. With immunofluorescence, we found that Nav1.5 and Kir6.2 are spatially localized closer to each other at the ICD compared to the lateral membrane (*Figure 6—figure supplement 3*). However, we do not think that Na$^+$ channels directly interact with K$_{ATP}$ channels. Our STORM imaging data show that the median intermolecular clustering distance from Nav1.5 to closest Kir6.2 at the ICD region is 528 nm (*Figure 6—figure supplement 3*), which is greater than expected for direct molecular interactions. The distributions of the cluster area of Nav1.5 and Kir6.2 channel subunits can be well fitted with single exponential function (*Figure 6—figure supplement 3*), indicating a stochastic self-assembly process in the formation of Nav1.5 and Kir6.2 clusters, which later become attached to AnkG in the ICD, as in the case of cardiac Ca$_v$1.2 directed by BIN1 (*Sato et al., 2019*). This observation suggests a model in which Na$^+$ channels and K$_{ATP}$ channels bind to closely spaced AnkG proteins. The suppression of Na$^+$ channel surface density by K$_{ATP}$ channel overexpression (and *vice versa*), which can be disrupted by mutagenesis of the Kir6.2 ankyrin binding motif, most likely results from competition of these two channels for the AnkG proteins. Several different proteins, including Nav channels, the Na$^+$/Ca$^{2+}$ exchanger, the Na$^+$/K$^+$ ATPase, IP3 receptors, KCNQ channels, and Kv3.1 channels all bind to ankyrins, but each protein class has a very different ankyrin binding site, which is evolutionarily highly conserved within the class (*Bennett and Healy, 2009*). The high sequence similarity of the ankyrin binding motifs of Kir6.2 and Nav1.5 (*Figure 3A*) is therefore somewhat surprising, but it is easy to visualize how these similar sequences can both bind to AnkG. More surprising, however, is that the corresponding sequence within Kir6.1 is almost identical, but neither supports ankyrin binding (*Li et al., 2010*), nor confers the ability to interact with Nav1.5 (*Figure 3B*). Future structural studies would be very helpful to determine the nature and specificity of the binding sites to AnkB and/or AnkG.

## Targeting of K$_{ATP}$ channels to lateral membranes

This study was not designed to study targeting to lateral membranes in cardiomyocytes, yet some of our findings are relevant. K$_{ATP}$ channels interact with AnkB in vitro (*Li et al., 2010*), and AnkB is expressed mainly at lateral membranes at Z- and M-lines (*Wu et al., 2015*). In support, when overexpressed in HEK293 cells, we found that Kir6.2 can interact both with AnkB and AnkG (*Figure 7—figure supplement 1*). We therefore fully expected a role for AnkB to target K$_{ATP}$ channels to lateral membranes. However, the peptide competition experiments question whether Kir6.2/AnkB interaction occurs natively. Peptides corresponding to ankyrin binding sites of Nav1.5 and Kir6.2 had identical effects to displace both Na$^+$ channels and K$_{ATP}$ channels from the ICD, demonstrating that the peptides were functional. However, the peptides did not influence these two channels at lateral membranes. Data with the duplex patch clamp technique suggest that (at least some of the) Na$^+$ channels and K$_{ATP}$ channels may be differently trafficked to the lateral membrane. Within the spatial dimension of duplex patch clamping, about half of the patches contained Na$^+$ channels at the lateral membrane, whereas K$_{ATP}$ channels were present in every single patch. Differential trafficking of these two channels in lateral membranes is also suggested from high-resolution scanning ion conductance microscopy (SICM), which demonstrated that the majority of Na$^+$ channels in lateral membranes are clustered in crests of mouse ventricular cardiomyocytes, with hardly any Na$^+$ channels present in the grooves (*Rivaud et al., 2017*). Early recordings with this technique, combined with whole-cell voltage clamping, by contrast suggested that lateral K$_{ATP}$ channels are present as submicrometer clusters in Z-grooves of the sarcolemma (*Korchev et al., 2000*). The finding is in support of the presence of K$_{ATP}$ channels in caveolae (*Yang et al., 2018*), which are often found at t-tubular structures (*Hong and Shaw, 2017*). Nevertheless, our duplex patch clamp data demonstrated functional co-localization of Na$^+$ channels an K$_{ATP}$ channels in about half of the patches and future studies should be directed at identification of the targeting mechanisms of these co-localized channels.

## What might be the physiological relevance and pathophysiological implications?

It is not clear what the role of $K_{ATP}$ channels at the ICD might be, and why they are functionally coupled to $Na^+$ channels. We have not examined action potential characteristics, since we believe that these local changes may not reflect global electrophysiological properties of the cell. Rather, we emphasize that our findings support the growing body of evidence that cardiac ion channels do not travel and organize as lone entities, but as complexes. Our data are fully supported by recent studies, such as the finding of co-translational 'microtranslatomes' that contain both $K^+$ channels and $Na^+$ channels (*Eichel et al., 2019*), and findings that $K^+$ and $Na^+$ channels can co-traffic in cardiac cells (*Ponce-Balbuena et al., 2018*). Given its small size (nanometers in scale) and convoluted nature, the ICD cleft space is severely diffusion restricted. Therefore, with repetitive electrical activity, $Na^+$ entering the cell via $Na^+$ channels may cause local intracellular $Na^+$ accumulation and $Na^+$ depletion in the ICD cleft, which is counteracted by ATP-driven $Na^+$ extrusion via the $Na^+/K^+$ pump in exchange for $K^+$ influx. This, in turn, may lead to $K^+$ depletion in the ICD cleft space. A $K^+$ flux coupling mechanism must exist to maintain homeostasis. In addition to their functional coupling to $Na^+$ channels (this study), $K_{ATP}$ channels and the $Na^+/K^+$ pump are also functionally coupled, such that an increased $Na^+/K^+$ pump activity activates $K_{ATP}$ channels (*Priebe et al., 1996*), which is thought to occur because of local sub-membrane ATP depletion (and ADP accumulation). We propose therefore that at high heart rates, the elevated $Na^+/K^+$ pump activity may locally activate the ICD $K_{ATP}$ channels in order to balance $K^+$ fluxes and maintain the ionic hemostasis in the ICD cleft (*Figure 8—figure supplement 1*). From genetic studies, the overwhelmingly predominant clinical phenotype of *KCNJ11* (Kir6.2) variants is insulin secreting disorders. Arrhythmias may well be a secondary and understudied phenotype. We know from pharmacological studies (with both humans and animals) that cardiac arrhythmias are a very real phenomenon associated with $K_{ATP}$ channel openers and blockers (*Foster and Coetzee, 2016*). In the model proposed, a relationship between $K_{ATP}$ channel activation and cardiac conduction is predicted, which is evident from the literature. For example, while studying $K_{ATP}$ channels in the cardiac specialized conduction system, we have observed that conduction slowing in ischemic Langendorff-perfused mouse hearts was essentially prevented by the $K_{ATP}$ channel blocker glibenclamide (*Bao et al., 2011*). This finding was in keeping with the literature that glibenclamide decreases conduction delays during ischemia in open-chest dogs (*Bekheit et al., 1990*), prevents the beneficial effect of IPC on electrical uncoupling during ischemia (*Tan et al., 1993*), and prevents asymmetric conduction slowing during acute ischemia in canine interventricular septum (*Morita et al., 2008*). Our data additionally demonstrate another level of pathophysiological relevance. We found that genetic variants associated with insulin disorders that cause $K_{ATP}$ channel trafficking defects may also affect $Na^+$ channel surface expression. These suggest that there may be implications for heart disease and conduction disorders in diabetic patients with $K_{ATP}$ channel trafficking mutations. Conversely, genetic defects resulting in $Na^+$ channel trafficking alterations have the potential to affect $K_{ATP}$ channel surface expression, and therefore the susceptibility of patients with inherited forms of arrhythmias to ischemia/reperfusion injury. These possible relationships need to be explored in future studies given the clinical relevance and therapeutic potential of our findings.

## Study limitations

At present, the physiological or pathophysiological implications of the functional interactions between $Na^+$ and $K_{ATP}$ channels are unknown. Human iPSC-derived cardiomyocytes are a poor substitute to study this question since these cells have an immature electrophysiology phenotype with little $K_{ATP}$ channel expression. These cells also lack fully developed intercalated disks. These limitations are shared by other cardiac cellular models, such as HL-1 cells and cultured primary neonatal cardiac myocytes. An in vivo model with disrupted AnkG/Nav1.5/$K_{ATP}$ interaction would be ideal to address effects in cardiomyopathies or in clinically relevant arrhythmias. Unfortunately no such model is currently available. Such an in vivo model would also be able to address the question of whether disrupted AnkG/Nav1.5/$K_{ATP}$ interaction affects the ICD structure. We deem this to be unlikely, though, given that patients with mutations in the Nav1.5 AnkG binding domain (and disrupted trafficking of Nav1.5 to the ICD) develop arrhythmias (Brugada syndrome) but not cardiomyopathies (*Mohler et al., 2004*), which would be expected to occur if structural disorder of the ICD occurred.

# Materials and methods

## Key resources table

| Reagent type (species) or resource | Designation | Source or reference | Identifiers | Additional information |
|---|---|---|---|---|
| Cell line (*Homo-sapiens*) | HEK293 | ATCC | Cat# CRL-1573, RRID:CVCL_0045 | Mycoplasma contamination negative |
| Recombinant DNA reagent | Nav1.5 | *Tan et al., 2018* | | |
| Recombinant DNA reagent | Kir6.2 | *Yang et al., 2018* | | |
| Recombinant DNA reagent | Kir6.2-myc | *Yang et al., 2018* | | |
| Recombinant DNA reagent | Avi-Kir6.2-myc | *Yang et al., 2018* | | |
| Recombinant DNA reagent | Kir6.2-KKK | This paper | Genscript | |
| Recombinant DNA reagent | Kir6.2-H259R | This paper | Genscript | |
| Recombinant DNA reagent | Kir6.1 | This paper | Genscript | |
| Recombinant DNA reagent | Kir6.2-AAA | *Tong et al., 2006* | | |
| Recombinant DNA reagent | SUR2A | *Yang et al., 2018* | | |
| Recombinant DNA reagent | Ankyrin-B | Addgene | RRID:Addgene_31057 | |
| Recombinant DNA reagent | Ankyrin-G | Addgene | RRID:Addgene_31059 | |
| Transfected construct | Adenovirus mCherry | This paper | Vector Biolabs | |
| Transfected construct (human) | Adenovirus Kir6.2-mEos3.2 | This paper | Vector Biolabs | |
| Peptide, recombinant protein | Kir6.2 ankyrin binding motif | Genscript | | VPIVAEEDGGGGGGRKKRRQRRRPQ |
| Peptide, recombinant protein | Nav1.5 ankyrin binding motif | Genscript | | VPIAVAESDGGGGGGRKKRRQRRRPQ |
| Antibody | Anti-Nav1.5 (Mouse monoclonal) | Sigma Aldrich | Cat# S8809, RRID:AB_477552 | WB (1:2000) |
| Antibody | Anti-Nav1.5 (Rabbit polyclonal) | Sigma Aldrich | Cat# S0819, RRID:AB_261927 | IF(1:200), STORM (1:50) |
| Antibody | Anti-Kir6.2 (Goat polyclonal) | Santa Cruz | Cat# sc-11226, RRID:AB_2130475 | WB (1:500) |
| Antibody | Anti-Kir6.2 (Chicken polyclonal) | *Hong et al., 2012* | C62 | IF(1:50), STORM (1:50) |
| Antibody | Anti-Kir6.2 (Rabbit polyclonal) | *Hong et al., 2012* | Lee62 | STORM (1:50) |
| Antibody | Anti-AnkyrinG (Mouse monoclonal) | Neuromab | Cat# N106/20, RRID:AB_2750699 | IF(1:500), STORM (1:50), WB (1:2000) |
| Antibody | Anti-AnkyrinB (Mouse monoclonal) | Neuromab | N105/17 | IF(1:500), STORM (1:50), WB (1:2000) |
| Antibody | Anti-myc (Mouse monoclonal) | Sigma Aldrich | 9E10 | WB (1:6000) |
| Antibody | Anti-GAPDH (Mouse monoclonal) | Sigma Aldrich | Cat# G8795, RRID:AB_1078991 | WB (1:20000) |

*Continued on next page*

*Continued*

| Reagent type (species) or resource | Designation | Source or reference | Identifiers | Additional information |
|---|---|---|---|---|
| Antibody | Anti-caveolin3 (Mouse monoclonal) | Transduction Laboratories | C38320 | WB (1:50000) |
| Antibody | donkey anti-mouse-HRP | Jackson ImmunoResearch | Cat# 715-035-150, RRID:AB_2340770 | WB (1:10000) |
| Antibody | donkey anti-goat-HRP | Jackson ImmunoResearch | Cat# 705-035-147, RRID:AB_2313587 | WB (1:10000) |
| Antibody | goat anti-chicken Alexa Fluor568 | Thermo Scientific | Cat# A-11041, RRID:AB_2534098 | IF(1:200) |
| Antibody | donkey anti-rabbit Alexa Fluor488 | Jackson ImmunoResearch | Cat# 711-545-152, RRID:AB_2313584 | IF(1:200) |
| Antibody | donkey anti-mouse Cy3 | Jackson ImmunoResearch | Cat# 715-165-151, RRID:AB_2315777 | IF(1:200) |
| Software, algorithm | GraphPad Prism | GraphPad Prism | RRID:SCR_002798 | |

## cDNA constructs and mutagenesis

Wild-type Nav1.5, Kir6.2, SUR2A constructs, Kir6.2-AAA, Kir6.2-myc and Avi-Kir6.2 cDNAs were previously used (*Tong et al., 2006*; *Yang et al., 2018*; *Tan et al., 2018*). Kir6.1, Kir6.2-KKK and Kir6.2-H259R were synthesized by Genscript. Ankyrin-G (#31059) and ankyrin-B (#31057) constructs were from AddGene.

## Cell culture and transfection

HEK-293 cells (ATCC CRL-1573), negative in mycoplasma contamination test, were cultured in EMEM with 10% fetal bovine serum. Lipofectamine 2000 (ThermoFisher, Waltham, MA) was used to transfect ankyrins, Nav1.5 and $K_{ATP}$ channel subunit cDNAs. When cells were transfected with multiple cDNAs, the empty vector (pcDNA3) was included to keep the total cDNA amount equal in transfection reactions. Nav1.5 and $K_{ATP}$ channels were co-transfected with the cDNA amount ratio of Nav1.5:Kir6.2:SUR2A to be 10:1:9. Rat ventricular cardiomyocytes were enzymatically isolated as previously described (*Hong et al., 2011*). All procedures conformed to the Guide for Care and Use of Laboratory Animals of the National Institutes of Health and were approved by the NYU IACUC committee (protocol s17-00352). Cells were plated on laminin-coated coverslips and cultured in EMEM. Adenovirus carrying mEos3.2 labeled Kir6.2 (Vector Biolabs, Malvern, PA, USA) were added at an MOI of 1000 for 12 hr incubation. An mCherry expressing adenovirus was used as control. Cultured cardiomyocytes were used for experiments 72 hr post-infection. Peptides corresponding to the ankyrin binding sites of Kir6.2 (VPIVAEEDGGGGGRKKRRQRRRPQ) or Nav1.5 (VPIAVAE SDGGGGGRKKRRQRRRPQ) were synthesized by Genscript, and incubated cardiomyocytes for 24 hr before patch clamp.

## Patch clamp electrophysiology

Standard patch-clamping was performed using an Axopatch-200B amplifier and recording data with a Digidata 1550A and Clampex 10 software. For inside-out $K_{ATP}$ current recordings, the pipette resistance was 3 ~ 4 MΩ when filled with pipette solution consisting of (in mM): 110 potassium gluconate, 30 KCl, 2 CaCl$_2$, 1 MgCl$_2$, 10 HEPES, and pH 7.4. The bath solution consisted of (in mM): 110 potassium gluconate, 30 KCl, 1 EGTA, 1 MgCl$_2$, 10 HEPES, and pH 7.2. Following patch excision, the pipette potential was held at +80 mV and current was digitized at 1 kHz. Currents were recorded immediately after patch excision and recordings with any sign of rundown were discarded. The 'cytosolic' ATP concentration was changed by a rapid solution changer (RSC160, BioLogic SAS, Seyssinet-Pariset, France). For whole-cell Nav1.5 current recordings in HEK293 cells, the pipette resistance was 2 ~ 3 MΩ when filled with pipette solution consisting of (in mM): 50 CsCl, 60 CsF, 10 TEA·Cl, 20 EGTA, 5 Na$_2$ATP, 10 HEPES, and pH 7.2 with CsOH. The bath solution consisted of (in mM): 30 NaCl, 110 CsCl, 4 KCl, 1 CaCl$_2$, 1 MgCl$_2$, 10 HEPES, 5 Glucose and pH 7.35 with CsOH. NaCl in bath solution dropped to 5 mM and compensated by CsCl when recording in cardiomyocytes.

For duplex recording of $Na^+$ and $K_{ATP}$ current from the same patch in cardiomyocytes, the pipette resistance was ~2 MΩ when filled with pipette solution consisting of (in mM): 125 NaCl, 5.4 KCl, 10 TEA·Cl, 1 $MgCl_2$, 0.33 $NaH_2PO_4$, 1 4-aminopyridine, 10 HEPES, and pH 7.35 with NaOH. The bath solution consisted of (in mM): 140 KCl, 0.33 $NaH_2PO_4$, 1 EGTA, 1 $MgCl_2$, 10 HEPES and pH 7.2 with KOH. Nav1.5 current was first measured by a voltage step protocol with P/N substraction in cell-attached mode, Nav1.5 channel was activated by depolarizations from −100 mV to voltages between −80 mV to +80 mV, then membrane under the pipette tip was excised and $K_{ATP}$ current was measured with rapid ATP perfusion in inside-out mode with holding potential of +80 mV.

## Electrophysiological recordings with angle SICM

Scanning ion conductance microscopy (SICM) is a non-contact scanning probe microscopy technique based on the principle that the flow of ions through the tip of a nanopipette filled with electrolytes decreases when the pipette approaches the surface of the sample (*Hansma et al., 1989*; *Korchev et al., 1997a*; *Korchev et al., 1997b*). In this study, we used a variant of SICM called angular approach scanning ion conductance microscopy described in detail by *Shevchuk et al. (2016)*. The system was used in the same configuration as *Leo-Macias et al. (2016)*. Briefly, the scanning probe was mounted in a PatchStar micromanipulator (Scientifica, UK) that allows to adjust the angle for the scanning, selected as 33° in this work for the purpose of scanning the ICD of adult cardiomyocytes with nanoscale resolution. Borosilicate glass nanopipettes pulled from 1.0 mm outer diameter, 0.4 mm ID capillary were used in all experiments. Axopatch 200B patch clamp amplifier (Axon Instruments; Molecular Devices) was used to measure the pipette current as well as to record ion channel activity. Cell-attached currents were digitized using Digidata 1440A and a pClamp 10 data acquisition system (Axon Instruments; Molecular Devices).

After the ICD region was recognized by the scanning, the pipette was moved to the area of sealing, the feedback of the hoping mode was switched off and a gigaseal was formed by lowering the pipette until it makes contact with the surface of the ICD. Cell-attached patch-clamp configuration was then used to record of $Na^+$ and $K_{ATP}$ channels simultaneously. Recordings were performed at room temperature using the following solutions; external solution containing (in mM): 145 KCl; 1 $MgCl_2$; 1 $CaCl_2$; 2 EGTA; 10 glucose; 10 HEPES; and pH 7.4 with KOH; internal recording solution containing (in mM): 135 NaCl; 0.4 $NaH_2PO_4$; 1 $MgCl_2$; 5.4 KCl; 1 $CaCl_2$; 5.5 glucose; 5 HEPES; 20 TEA-Cl; 0.2 $CdCl_2$; 10 CsCl; 10, 4-AP; and pH 7.4 with NaOH. Pinacidil was added to both solutions at a concentration of 200 μM to activate $K_{ATP}$ channels. The pipette used for cell-attached recordings had an average resistance of ~30 MΩ. To generate a current–voltage (I–V) relationship that allow the simultaneous recording of $Na^+$ and $K_{ATP}$ channels, the membrane under the patch was held at a voltage of −120 mV and incremental steps of 10 mV were applied from −100 to +90 mV. Data were low-pass filtered at 1 kHz using the built-in Bessel filter of the amplifier and sampled at 20 kHz.

## Biotinylation assay

HEK293 cells expressing Nav1.5 and Avi-Kir6.2-myc were incubated with 0.33 mM biotin for 1 hr at 4°C. After washing with PBS, cells were homogenized in RIPA buffer. Equal amount of biotinylated proteins was incubated with Neutravidin agarose beads (Thermo Scientific) at 4°C overnight. The supernatants were discarded and biotinylated proteins were eluted by a mixture of loading buffer and 200 mM DTT. Western blots were quantified by ImageJ.

## Membrane fractionation

Flash-frozen hearts were ground to a fine powder in liquid nitrogen using a pestle and mortar. Samples were homogenized on ice with 30 strokes of a glass-glass homogenizer, followed by 30 strokes in a Dounce homogenizer in (in mM) 250 sucrose, 1 EDTA, 10 HEPES, 1 DTT and pH 7.4 supplemented with protease inhibitor cocktail (Roche Applied Science). Following brief centrifugation (1000 g for 5 min at 4°C), the pellet was re-homogenized in fresh homogenization buffer with 25 strokes of a tight-fitting Dounce and cleared by brief centrifugation (1000 g, 5 min, 4°C). The resulting supernatant was combined with that of the previous step. The supernatant was centrifuged at

50,000 rpm using a 90 Ti rotor (Beckman Coulter, Brea, CA) for 1 hr at 4°C. The resulting membrane pellets were solubilized with rotation overnight at 4°C in 20 mM HEPES, 0.5% Triton X100, pH 7.4.

## Immunocytochemistry and immunohistochemistry

As previously described (*Yang et al., 2018*), isolated rat cardiomyocytes or rat heart slices were fixed with 4% paraformaldehyde. Cells or tissue were permeabilized with 0.1% Triton X-100 and blocked with 5% donkey serum in PBS. Primary and secondary antibodies buffered in blocking solution were sequentially applied. After washing and mounting, images were obtained by Zeiss 700 confocal microscope (Zeiss, Jena, Germany).

## Stochastic optical reconstruction microscopy (STORM)

Freshly isolated rat cardiomyocytes were plated on laminin-coated coverslips for 1 hr before fixation with 4% paraformaldehyde. Cells were then permeabilized with 0.1% Triton in PBS for 10 min, and incubated in blocking solution (PBS based 2% Glycine, 2% BSA and 0.2% Gelatin) for 30 min. Primary antibodies diluted 1:50 in blocking solution incubated the cells for 1 hr at room temperature. After three washes with PBS, secondary antibodies against a combination of mouse conjugated with Alexa Fluor 647 and rabbit conjugated with Alexa Fluor 568 or a combination of rabbit conjugated with Alexa Fluor 647 and chicken conjugated with Alexa Fluor 568 (1:10000, Invitrogen) were incubated for 15 min at room temperature. Imaging conditions were achieved by addition of 200 mmol/L mercaptoethylamine and an oxygen scavenging system (0.4 mg/ml glucose oxidase, 0.8 µg/mL catalase and 10% (wt/wt) glucose) to the fluorophore-containing sample.

As previously described (*Kim et al., 2019*), samples were imaged using a custom-built platform based on an inverse microscopy setup (Leica DMI3000). Sample emission was split into two channels through proper dichroic and emission narrow-band bandpass filters (green channel 580/60, red channel 680/40, Semrock), in conjunction with a Dual View (DV2-Photometrics) to image two colors simultaneously, onto a single EM-CCD camera (Andor iXon+ 897). For accurate alignment and mapping of the two color channels, we first imaged diffraction-limited fluorescent beads that have a wide emission spectra spanning both channels (Invitrogen). The location of the beads was matched for both channels based on the use of a polynomial morph-type mapping function, whereby mapping coefficients are generated by Gaussian and centroid fits to the sub-diffraction limit point-spread functions of the fluorescence beads. The two-color image was reconstructed at 20 nm/pixel using the following QuickPALM parameters, FWHM = 4 and S/N = 2.00. The reconstructed super-resolved images of each channel were then super-imposed to generate a two color super-resolved image. The mapping error in the super-resolved image was 20 nm. ROIs of intercalated disc were manually drawn for each reconstructed super-resolution images and further cluster detection were obtained using ImageJ and cluster distance analysis accomplished by a home-built Python script that utilized the image processing packages scikit-image (*van der Walt et al., 2014*), and 'Mahotas,' an open source software for scriptable computer vision (http://dx.doi.org/10.5334/jors.ac). Please refer to our previous studies (*Agullo-Pascual et al., 2013*) for extensive methodological details of our custom-developed analysis pipeline, its technical specifications and limitations, as well as specifics of its resolution both in the X-Y and in the Z planes.

## Antibodies

Primary antibodies used were: mouse anti-GAPDH (G8795, Sigma-Aldrich; 1:10000 for western blot), mouse anti-c-Myc (M4439, Sigma-Aldrich; 1:6000 for western blot), mouse anti-Nav1.5 (S8809, Sigma-Aldrich; 1:2000 for western blot), rabbit anti-Nav1.5 (S0819, Sigma-Aldrich; 1:200 for immunostaining, 1:50 for STORM), chicken anti-Kir6.2 (C62; 1:50 for immunostaining and STORM), rabbit anti-Kir6.2 (Lee62; 1:50 for STORM), goat anti-Kir6.2 (N18, Santa Cruz; 1:500 for western blot), mouse anti-ankyrin-B (105/17, Neuromab; 1:2000 for western blot, 1:50 for STORM) and mouse anti-ankyrin-G (106/20, Neuromab; 1:2000 for western, 1:50 for STORM). Secondary antibodies used were donkey anti-mouse-HRP (715-035-150, Jackson, 1:10000), donkey anti-goat-HRP (705-035-147, Jackson, 1:10000), goat anti-chicken Alexa Fluor568 (A-11041, Thermo Scientific; 1:200), donkey anti-rabbit Alexa Fluor488 (711-545-152, Jackson; 1:200), and donkey anti-mouse Cy3 (715-165-151, Jackson; 1:200). All antibodies used in this study have been fully validated, either experimentally or in the literature. Details can be found in *Figure 6—figure supplement 1* and *Supplementary file 1*.

## Statistical analysis

The sample size was determined using power analysis. The number of biological replicates are indicated in the figure legends. Single random sampling was used for all experiments. When comparing two groups, we used the Student's t-test. A one-way or two-way ANOVA was used for comparison of multiple groups, followed by the Tukey's post-hoc analysis for comparisons to a single control. A value of $p < 0.05$ was considered significant.

## Acknowledgements

This work was supported by R01 HL126905 (WAC), R01 HL146514 (WAC), RO1 HL134328, RO1 HL136179, RO1 HL145911 and a Fondation Leducq Transatlantic Network (MD), an Excellence Scholarship from the Rafael del Pino Foundation (MP-H), BBSRC BB/M022080/1 (AS), ROI HL126802 (JG) British Heart Foundation RG/17/13/33173 (JG and JSA), and an AHA Postdoctoral Fellowship award 17POST33370050 (HQY).

## Additional information

### Funding

| Funder | Grant reference number | Author |
| --- | --- | --- |
| National Institutes of Health | HL126905 | William A Coetzee |
| Fondation Leducq | | Eli Rothenberg<br>Mario Delmar |
| Rafael del Pino Foundation | | Marta Pérez-Hernández |
| Biotechnology and Biological Sciences Research Council | BB/M022080 | Andriy Shevchuk |
| British Heart Foundation | RG/17/13/33173 | Jose Sanchez-Alonso<br>Julia Gorelik |
| American Heart Association | 17POST33370050 | Hua-Qian Yang |
| National Institutes of Health | HL146514 | William A Coetzee |
| National Institutes of Health | HL134328 | Mario Delmar |
| National Institutes of Health | HL136179 | Mario Delmar |
| National Institutes of Health | HL145911 | Mario Delmar |
| National Institutes of Health | HL126802 | Julia Gorelik |

The funders had no role in study design, data collection and interpretation, or the decision to submit the work for publication.

### Author contributions

Hua-Qian Yang, Conceptualization, Resources, Data curation, Software, Formal analysis, Validation, Investigation, Visualization, Methodology; Marta Pérez-Hernández, Software, Formal analysis, Validation, Investigation, Visualization, Methodology; Jose Sanchez-Alonso, Formal analysis, Investigation, Methodology; Andriy Shevchuk, Eli Rothenberg, Data curation, Software, Formal analysis, Investigation, Methodology; Julia Gorelik, Conceptualization, Data curation, Software, Formal analysis, Supervision, Funding acquisition, Investigation, Methodology; Mario Delmar, Conceptualization, Resources, Supervision, Investigation, Methodology; William A Coetzee, Conceptualization, Resources, Data curation, Formal analysis, Supervision, Funding acquisition, Validation, Investigation, Methodology, Project administration

### Author ORCIDs

Hua-Qian Yang (iD) https://orcid.org/0000-0002-9402-6222
Julia Gorelik (iD) http://orcid.org/0000-0003-1148-9158

Eli Rothenberg [iD] http://orcid.org/0000-0002-1382-1380
William A Coetzee [iD] https://orcid.org/0000-0003-1522-8326

## Ethics

Animal experimentation: This study was performed in strict accordance with the recommendations in the Guide for the Care and Use of Laboratory Animals of New York University School of Medicine (protocol s17-00352).

## Decision letter and Author response

Decision letter https://doi.org/10.7554/eLife.52373.sa1
Author response https://doi.org/10.7554/eLife.52373.sa2

## Additional files

### Supplementary files

- Source data 1. Original data and graph files.
- Supplementary file 1. Validation of antibodies.
- Transparent reporting form

### Data availability

All data generated or analysed during this study are included in the manuscript and supporting files.

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
