## [Decision Letter]

**Acceptance summary:**

Emerging evidence in the cardiac ion channel field challenges the traditional idea that treats ion channels as isolated entities. The cardiac ICD is a complex protein network, which contains ion channels and structural proteins. Its function is to maintain mechanical and electrical coupling among neighboring cardiomyocytes in the tridimentional myocardium. AnkG plays a role in Nav1.5 organization (Lowe et al., 2008; Leterrier et al., 2014) and in this manuscript, Yang provides additional evidence for NaV1.5 interactions with Kir6.2 mediated by Ankyrin-G, but not Ankyrin-B, in the cardiomyocyte. The authors investigate Nav1.5 and K_ATP_ channel distribution at the intercalated disc (ICD) and their reciprocal interaction mediated by the intracellular structural platforms. They conducted a functional analysis of such interactions using two models: transfected HEK293 cells (5 cell groups: Na^+^ K_ATP_ channels, Na+, K_ATP_, and empty) and isolated rat cardiomyocytes. Their data suggests that Na^+^ and K_ATP_ channels are paired-located at the ICD and have a functional interaction that is mediated by the anchoring protein Ankirin-G (AnkG).

**Decision letter after peer review:**

Thank you for submitting your article "Ankyrin-G mediates targeting of both Na^+^ and K_ATP_ channels to the cardiac intercalated disc" for consideration by *eLife*. Your article has been reviewed by three peer reviewers, including Baron Chanda as the Reviewing Editor and Reviewer #1, and the evaluation has been overseen by Kenton Swartz as the Senior Editor.

The reviewers have discussed the reviews with one another and the Reviewing Editor has drafted this decision to help you prepare a revised submission.

Essential revisions:

1) While two different models were used, none of them accurately illustrates human physiology. It would be of interest to use hiPSC-CM or a genetically modified mouse model to address the aim of this manuscript. Furthermore, it is well known that the integrity of the ICD components is important for normal heart function and structure. An in vivo model of this AnkG-Nav1.5-K_ATP_ disruption could address its implication in cardiomyopathy or clinically relevant arrhythmia development. We ask that the authors clearly discuss the limitations of their current approach in the Discussion.

2) Related to the above point – Previous literature shows that the trafficking machinery in the HEK cells exhibits significant differences compared to cardiomyocytes (Nav1.5 E1053K in HEK cells transports to membrane but does not in the rat ventricular cardiomyocytes- Mohler et al., 2004). The authors should discuss the caveats of their findings which are primarily obtained using heterologous expression system.

3) It has been demonstrated that Ank-G and Cx43 are necessary to preserve INA amplitude, electrical coupling and intercellular adhesion strength (Sato et al., 2011). The authors should clarify the possible role of Cx43 in the protein-protein interactions they are studying: can the authors demonstrate a Cx43-independent relation between K_ATP_, Na^+^ and AnK-G interaction in the cardiomyocyte? In other words, is Cx43 affected in rat ventricular cardiomyocytes?

4) The suggested impaired Na+, K_ATP_ and Ank-G macromolecular complex may lead to disruption of the ICD organization. This would lead to functional consequences on proteins, such as plakophilin-2, which regulates the function of ion channels responsible for the action potential. Is there any evidence for structural disorder of the ICD when the Nav15-K_ATP_-AnkG complex is disrupted? Can the authors provide information on the consequences of the above interactions on action potential characteristics in rat ventricular cardiomyocytes?

5) The KCNJ11 gene encodes the inwardly rectifying potassium channel Kir6.2, which has been related mainly endocrine disorders, including diabetes mellitus. To the knowledge of this reviewer, there is no previous evidence suggesting that Kir6.2 is involved in cardiac arrhythmic disorders. The authors must be aware of the limited clinical implications of their data in terms of patient management and therapeutic options.

6) Isabelle Deschenes' lab has described quite convincingly that Nav1.5 channels form dimers that gate cooperatively. I am surprised her papers are not cited in this study and discussed in the context of the current findings.

7) The authors must include histograms of the cluster area distributions of K_ATP_ and Nav1.5 channels. Should the distributions be exponential, the authors should at least discuss the possibility of stochastic self-assembly mechanism for cluster formation. Sato et al., 2019 is a good starting point.

8) Do K_ATP_ and Nav1.5 negatively regulate their cluster size? What ratios of cDNA (Nav 1.5 to K_ATP_) were used to transfect the HEK cells? Is there a particular ratio of the channels that has to be transfected to see the functional interaction? This part is not clear from the Materials and methods section.

Following up on that, the Materials and methods section for heterologous expression does not state whether that Ankyrin-G was transfected along the two ion channels. It is not clear why AnkG was specifically transfected for the coimmunoprecipitation experiments (Figure 6—figure supplement 3). Please clarify.

9) The authors should discuss in great detail validation of Abs.

10) The authors should provide nano ruler data to validate the lateral resolution of their STORM system. Pritchard et al., 2012 PNAS provides guidance on how to do that.

11) The authors mention that the Kir6.2/KKK mutant does not functionally interact with the Nav 1.5, however the immunoblots shown in Figure 3—figure supplement 1 shows a decrease in the Nav1.5 expression with increasing amounts of K_ATP_ 6.2-KKK cDNA transfection. This needs to be clarified. Was total cell lysate used to the experiment in Figure 3—figure supplement 1 and do the authors see a similar decrease on the surface expression as well?

12) The co-IP experiment in Figure 7—figure supplement 1 is an important piece of work considering that the authors claim that a common scaffolding protein underlies the functional interaction. Prior research does show that Kir6.2 binds to AnkB, do the authors think that the binding of Kir6.2 to AnkG also occurs through the same binding Ankyrin binding site? Please clarify.

---

## [Author Response]

Essential revisions:1) While two different models were used, none of them accurately illustrates human physiology. It would be of interest to use hiPSC-CM or a genetically modified mouse model to address the aim of this manuscript. Furthermore, it is well known that the integrity of the ICD components is important for normal heart function and structure. An in vivo model of this AnkG-Nav1.5-K_ATP_ disruption could address its implication in cardiomyopathy or clinically relevant arrhythmia development. We ask that the authors clearly discuss the limitations of their current approach in the Discussion.

We fully agree that an in vivo model of the AnkG/Nav1.5/K_ATP_ disruption would address its implication in cardiomyopathy or clinically relevant arrhythmia development. Unfortunately no such model is currently available. hiPSC cardiomyocytes are a poor substitute since in our experience these cells have an immature electrophysiology phenotype with little or no K_ATP_ channel expression. These cells also lack t-tubules and fully developed intercalated disks. These limitations are shared by other cardiac cellular models, such as HL-1 cells and cultured primary neonatal cardiac myocytes. A new section (“Study Limitations”) has been added to clearly discuss these limitations in the Discussion section.

2) Related to the above point – Previous literature shows that the trafficking machinery in the HEK cells exhibits significant differences compared to cardiomyocytes (Nav1.5 E1053K in HEK cells transports to membrane but does not in the rat ventricular cardiomyocytes- Mohler et al., 2004). The authors should discuss the caveats of their findings which are primarily obtained using heterologous expression system.

Our data with peptides against Ank binding sites suggest that the trafficking events that we detect in HEK293 cells also occur in isolated cardiomyocytes (Figure 7). Nevertheless, this is an excellent point that is now discussed.

3) It has been demonstrated that Ank-G and Cx43 are necessary to preserve INA amplitude, electrical coupling and intercellular adhesion strength (Sato et al., 2011). The authors should clarify the possible role of Cx43 in the protein-protein interactions they are studying: can the authors demonstrate a Cx43-independent relation between K_ATP_, Na^+^ and AnK-G interaction in the cardiomyocyte? In other words, is Cx43 affected in rat ventricular cardiomyocytes?

Since the K_ATP_, Na^+^ and AnK interaction can be demonstrated in HEK-293 cells, which do not express Cx43, our data demonstrate that Cx43 is not a necessary component for functional interaction. However, we have not investigated this in cardiomyocytes. Current experimental models do not allow us to directly investigate this question. The peptide experiments require isolated cardiomyocytes and it is well documented that Cx43 is rapidly internalized from the ICD following enzymatic cell isolation. An in vivo model of the AnkG/Nav1.5/K_ATP_ disruption may address this question, but such a model is not currently available. We have addressed this limitation in the revised Discussion.

4) The suggested impaired Na+, K_ATP_ and Ank-G macromolecular complex may lead to disruption of the ICD organization. This would lead to functional consequences on proteins, such as plakophilin-2, which regulates the function of ion channels responsible for the action potential. Is there any evidence for structural disorder of the ICD when the Nav15-K_ATP_-AnkG complex is disrupted? Can the authors provide information on the consequences of the above interactions on action potential characteristics in rat ventricular cardiomyocytes?

Dr. Mohler’s laboratory has demonstrated that patients with mutations in the Nav1.5 AnkG binding domain (and disrupted trafficking of Nav1.5 to the ICD) develop arrhythmias (Brugada syndrome) but not cardiomyopathies. The latter would be expected to occur if structural disorder of the ICD occurred. Therefore, we deem it unlikely that disruption of Nav1.5/K_ATP_ localization to the ICD would result in structural disorder of the ICD. This issue is now discussed. We have not examined action potential characteristics, since we believe that these are local changes that may not reflect global electrophysiological properties of the cell. Rather, we emphasize that our findings support the growing body of evidence that cardiac ion channels do not travel and organize as lone entities, but as complexes. Our data are fully supported by recent studies, such as Gail Robertson’s finding of co-translational 'microtranslatomes' that contain both K channels and Na channels, and Pepe Jalife’s finding that K^+^ and Na^+^ channels can co-traffic in cells. We have made a better attempt in the revised manuscript to convey this message.

5) The KCNJ11 gene encodes the inwardly rectifying potassium channel Kir6.2, which has been related mainly endocrine disorders, including diabetes mellitus. To the knowledge of this reviewer, there is no previous evidence suggesting that Kir6.2 is involved in cardiac arrhythmic disorders. The authors must be aware of the limited clinical implications of their data in terms of patient management and therapeutic options.

Genetic studies are not necessarily the *de facto* standard for an involvement in arrhythmias. For example, a recent report of the NIH Clinical Genome Resource Consortium concluded that, when performing a systematic evaluation of the evidence supporting the causality of gene variants associated with Brugada syndrome, clinical validity was demonstrated for only one gene (*SCN5A*), even though over 20 genes have previously been implicated with Brugada syndrome [PMID:29959160]. In the case of *KCNJ11*, the overwhelmingly predominant clinical phenotype is insulin secreting disorders and arrhythmias may well be a secondary understudied phenotype. We know from pharmacological studies (with both humans and animals) that cardiac arrhythmias are a very real phenomenon associated with the use of K_ATP_ channel openers and blockers. We are therefore very aware of the potential clinical implications of our data in terms of patient management and therapeutic options. We have expanded this topic in the revised manuscript.

6) Isabelle Deschenes' lab has described quite convincingly that Nav1.5 channels form dimers that gate cooperatively. I am surprised her papers are not cited in this study and discussed in the context of the current findings.

With all due respect and at the risk of sounding ignorant, we are unclear as to the reason behind the reviewer’s surprise. The fact that sodium channels cluster together was actually first demonstrated in adult cardiac myocytes by the Delmar lab (PMID:26787348). That the channels gate cooperatively, though beautifully demonstrated by the paper of the Deschenes lab, seems outside of the realm of relevance for the present manuscript. Yet, if the reviewer considers this to be of critical importance, and if he/she were kind enough to provide us a rationale for inclusion of this paper in our Discussion, we would be happy to oblige.

7) The authors must include histograms of the cluster area distributions of K_ATP_ and Nav1.5 channels. Should the distributions be exponential, the authors should at least discuss the possibility of stochastic self-assembly mechanism for cluster formation. Sato et al., 2019 is a good starting point.

We now show these data as Figure 6—figure supplement 3, and have discussed it in the revised manuscript. We apologize for the omission of these important data.

8) Do K_ATP_ and Nav1.5 negatively regulate their cluster size? What ratios of cDNA (Nav 1.5 to K_ATP_) were used to transfect the HEK cells? Is there a particular ratio of the channels that has to be transfected to see the functional interaction? This part is not clear from the Materials and mmethods section.Following up on that, the Materials and methods section for heterologous expression does not state whether that Ankyrin-G was transfected along the two ion channels. It is not clear why AnkG was specifically transfected for the coimmunoprecipitation experiments (Figure 6—figure supplement 3). Please clarify.

The cDNA ratios used are now clearly indicated in the revised Materials and methods section and/or Figure legends. We have not been sufficiently clear to point out that we have not transfected cells with AnkB or AnkG, with the exception of testing the validity of the antibodies and in the co-IP experiment (Figure 7—figure supplement 1). In all other cases, we have relied on the endogenous expression of Ankyrins in HEK-293 cells or in cardiomyocytes.

9) The authors should discuss in great detail validation of Abs.

The details of antibody validation are now added as a supplement file (Supplementary file 1). Antibodies used in HEK293 cells are validated by transfected vs. untransfected cell samples (Figure 2, Figure 3—figure supplement 1 and Figure 7—figure supplement 1). Antibodies used in cardiomyocytes are validated by knockout tissue samples either in this paper (Figure 6—figure supplement 1) or from the literature.

10) The authors should provide nano ruler data to validate the lateral resolution of their STORM system. Pritchard et al, 2012 PNAS provides guidance on how to do that.

We thank the reviewer for this suggestion. Unfortunately, we are unable to find the article that the reviewer would like us to consult. PubMed does not list any article authored by Pritchard et al. that is published in PNAS in 2012 and addresses lateral resolution of STORM. That being said, the point of the reviewer (namely, to provide validation of the lateral resolution of our STORM system), is well taken. In this regard, the reviewer is invited to consult the previous work from the Rothenberg/Delmar labs (a total of 12 papers, starting in 2012; the first one in close collaboration with Dr. Coetzee) in which we have applied STORM methods. Of particular relevance to this conversation is the paper by Agullo-Pascual et al., published in Cardiovascular Research 2013 (PMID:23929525). In the Supplemental material of that paper we provide extensive methodological details of our custom-made analysis system, its technical specifications and limitations, as well as specifics of its resolution both in the X-Y and in the Z planes. In that paper (and in the technical supplement) we cite additional reference materials, including several papers of Dr. Rothenberg dealing with single-molecule localization microscopy, starting in 2010. We believe that the previous work provides enough details on the characterization of the technical aspects of our system, our methods of measurement and the resolution in the X-Y plane. We do agree that these details were not specified in the present manuscript and therefore we have included additional references and wording.

11) The authors mention that the Kir6.2/KKK mutant does not functionally interact with the Nav 1.5, however the immunoblots shown in Figure 3—figure supplement 1 shows a decrease in the Nav1.5 expression with increasing amounts of K_ATP_ 6.2-KKK cDNA transfection. This needs to be clarified. Was total cell lysate used to the experiment in Figure 3—figure supplement 1 and do the authors see a similar decrease on the surface expression as well?

The apparent decrease of Nav1.5 expression by Kir6.2-KKK is a result of the experimental conditions used. By raising the Kir6.2-KKK cDNA amount relative to that of Nav1.5, we believe that less Nav1.5 cDNA was taken up into the cell during transfection because there is more of the other cDNAs. Indeed, we have now performed an experiment which demonstrated that less Nav1.5 expression takes place even when increasing the overall amount of empty vector (pcDNA3) in the transfection reaction (Figure 3—figure supplement 1). This explanation now provided in the figure legend.

12) The co-IP experiment in Figure 7—figure supplement 1 is an important piece of work considering that the authors claim that a common scaffolding protein underlies the functional interaction. Prior research does show that Kir6.2 binds to AnkB, do the authors think that the binding of Kir6.2 to AnkG also occurs through the same binding Ankyrin binding site? Please clarify.

Our data demonstrate that peptides against a region of Kir6.2, previously identified as an AnkB binding site, and peptides against the Nav1.5 AnkG binding site have similar effects. They each affect expression of Na and K_ATP_ channels at the ICD, but not at the lateral membrane. Since AnkB is not readily detected at the ICD, our conclusion is that these peptides disrupt interaction with AnkG at the ICD. This is a plausible conclusion given the near sequence identity of these peptide sequences. We assume that the editorial panel refers to the binding site(s) within AnkB or AnkG. Unfortunately, we do not have information about the nature of these sites, not have we addressed these experimentally.